# Co-expression of xenopsin and rhabdomeric opsin in photoreceptors bearing microvilli and cilia

Oliver Vöcking[1,2], Ioannis Kourtesis[1], Sharat Chandra Tumu[1], Harald Hausen[1]*

[1]Sars International Centre for Marine Molecular Biology, University of Bergen, Bergen, Norway; [2]Department of Ophthalmology, University of Pittsburgh, Pittsburgh, United States

**Abstract** Ciliary and rhabdomeric opsins are employed by different kinds of photoreceptor cells, such as ciliary vertebrate rods and cones or protostome microvillar eye photoreceptors, that have specialized structures and molecular physiologies. We report unprecedented cellular co-expression of rhabdomeric opsin and a visual pigment of the recently described xenopsins in larval eyes of a mollusk. The photoreceptors bear both microvilli and cilia and express proteins that are orthologous to transporters in microvillar and ciliary opsin trafficking. Highly conserved but distinct gene structures suggest that xenopsins and ciliary opsins are of independent origin, irrespective of their mutually exclusive distribution in animals. Furthermore, we propose that frequent opsin gene loss had a large influence on the evolution, organization and function of brain and eye photoreceptor cells in bilaterian animals. The presence of xenopsin in eyes of even different design might be due to a common origin and initial employment of this protein in a highly plastic photoreceptor cell type of mixed microvillar/ciliary organization.

DOI: https://doi.org/10.7554/eLife.23435.001

## Introduction

Animal eyes are amongst the best-investigated sensory organs and are subject to studies not only in sensory biology, but also in general molecular, developmental and evolutionary biology. Comparative investigations yield insights into the diversity of light-detection mechanisms and their function across animals. Further, the integrative data sets on eyes and their cellular components has had considerable impact on the characterization of different kinds of light-detecting cells, and on concepts how to trace their evolution and that of the organs they constitute (*Arendt, 2003*; *Arendt et al., 2016*; *Oakley and Speiser, 2015*; *Wagner, 2014*). These studies have dealt with the general question of whether gradual change or rather integration and reshuffling of modules that also exist elsewhere is the main driving force in the evolution of light-detecting cell types.

Traditionally, photoreceptor cells (PRCs) have been characterized mainly by their general electrophysiological response and their fine structure, which reveal clear differences between the two main systems of interest, the eyes of insects and those of vertebrates. The PRCs in insects have been shown to enlarge the sensory area by microvilli and to depolarize in response to light, whereas vertebrate eyes are hyperpolarizing and bear a modified cilium (*Eakin, 1963*; *1968*, *1979*). These findings initiated a wealth of comparative studies and resulted in pro-longed debates on whether a few conserved lineages of animal eye PRCs exist or whether both those PRCs andwhole eyes evolved multiple times independently (*Eakin, 1982*; *von Salvini-Plawen, 1982*; *Vanfleteren, 1982*).

Subsequent data on molecular physiology and transcriptional regulation of cell specification allowed much more detailed characterization of PRCs,suggesting that two conserved PRC types are

**\*For correspondence:** harald.hausen@uib.no

**Competing interests:** The authors declare that no competing interests exist.

**eLife digest** Animal eyes have photoreceptor cells that contain light-sensitive molecules called opsins. Although all animal photoreceptor cells of this kind share a common origin, the cells found in different organisms can differ considerably. The photoreceptor cells in flies, squids and other invertebrates store a type of opsin called r-opsin in thin projections on the surface known as microvilli. On the other hand, the visual photoreceptor cells in human and other vertebrate eyes transport another type of opsin (known as c-opsin) into more prominent extensions called cilia.

It has been suggested that the fly and vertebrate photoreceptor cells represent clearly distinct evolutionary lineages of cells, which diverged early in animal evolution. However, several organisms that are more closely related to flies than to vertebrates have eye photoreceptor cells with cilia. Do all eye photoreceptors with cilia have a common origin in evolution or did they emerge independently in vertebrates and certain invertebrates?

The photoreceptor cells of a marine mollusc called *Leptochiton asellus*, are unusual because they bear both microvilli and cilia, suggesting they have intermediate characteristics between the two well-known types of photoreceptor cells. Previous studies have shown that these photoreceptor cells use r-opsin, but Vöcking et al. have now detected the presence of an additional opsin in the cells. This opsin is a member of the recently discovered xenopsin family of molecules. Further analyses support the findings of previous studies that suggested this type of opsin emerged early on in animal evolution, independently from c-opsin. Other invertebrates that have cilia on their eye photoreceptors also use xenopsin and not c-opsin.

The findings of Vöcking et al. suggest that, in addition to c-opsin and r-opsin, xenopsin has also driven the evolution of photoreceptor cells in animals. Eye photoreceptor cells in invertebrates with cilia probably share a common origin with the microvilli photoreceptor cells that is distinct from that of vertebrate visual cells. The observation that two very different types of opsin can be produced within a single cell suggests that the molecular processes that respond to light in photoreceptor cells may be much more complex than previously anticipated. Further work on these processes may help us to understand how animal eyes work and how they are affected by disease.
DOI: https://doi.org/10.7554/eLife.23435.002

indeed the main sensory components of animal cerebral eyes: (a) microvillar depolarizing r-opsin-expressing PRCs, which signal via the Gq-mediated IP3 cascade, and (b) ciliary hyperpolarizing c-opsin-expressing cells, which signal via the Gi/t-mediated cGMP cascade (*Arendt et al., 2002a, 2004*; *Fain et al., 2010*; *Fernald, 2006*; *Gehring, 2014*). This evolution-based classification of PRCs became a sound basis for many kinds of comparative eye research and for ideas about how animal eyes evolved.

The r-opsin+ type of PRCs constitutes the main visual elements in many protostome cerebral eyes, but they are also present in the ganglion cell layer of vertebrate eyes, where they are involved in the pupillary reflex and the entrainment of the circadian clock (*Hattar et al., 2003*; *Koyanagi et al., 2005*; *Panda et al., 2002*). Ciliary c-opsin+ PRCs on the other hand are predominantly known as the visual PRCs of vertebrate eyes and the frontal eye of cephalochordates (*Vopalensky et al., 2012*), while their role in protostomes is controversially discussed. In protostomes, c-opsins were initially reported only in arthropod and annelid deep brain photoreceptors (*Arendt et al., 2004*; *Velarde et al., 2005*). It was assumed, therefore, that this visual pigment was ancestrally employed by unpigmented brain PRCs that were fulfilling non-visual functions and were only secondarily integrated into the eyes of chordates (*Arendt et al., 2004*; *Vopalensky et al., 2012*). Reports on the presence of ciliary PRCs and c-opsins in the eyes of protostomes challenged this view and also the concept of gradually evolving ancient PRC types in animal eyes. Accordingly, a rather flexible employment of different opsin types by eye photoreceptor cells or frequent gain and loss of photoreceptor cell types in cerebral eyes during the evolution of bilaterian animals were suggested (*Passamaneck et al., 2011*).

Advances in opsin phylogeny, however, suggest that several protostome sequences, which share characteristic sequence motifs with c-opsins and were initially classified as such, fall into their own group, now termed xenopsins (*Ramirez et al., 2016*). In this situation, it is highly desirable to obtain

deeper insights into the evolutionary history of c-opsins and xenopsins and their employment in animal photoreceptor cells, especially in those of protostome invertebrates. By deeply mining public and new data, we find xenopsins to be present in several taxa of protostome invertebrates. Although we did not find even a single case of co-existence of c-opsins and xenopsins in any organism, our phylogenetic and gene structure analyses strongly suggest that xenopsins and c-opsins are two distinct opsin subgroups. A new xenopsin from *Leptochiton asellus*, a member of the basally branching mollusk group of chitons, is found to be co-expressed with r-opsin in PRCs bearing both microvilli and cilia. This raises interesting questions about the light transduction and molecular physiology of the cells and their evolutionary origin. Recompilation of existing and new data challenges the dichotomous distinction of bilaterian eye PRCs into ciliary c-opsin+ and microvillar r-opsin+ cell types, but nevertheless supports the view that conserved lineages of eye PRCs exist. We suggest that a highly plastic PRC type of mixed microvillar/ciliary organization and frequent lineage-specific loss of c-opsins and xenopsins were important players in animal eye evolution.

## Results

### A new xenopsin from the chiton *Leptochiton asellus*

By screening RNA-seq data of *Leptochiton asellus*, we detected a sequence showing clear opsin characteristics. This was substantiated by reciprocal BLAST as well as domain and motif analyses. The sequence possesses the Pfam tm7_1 domain, Lys296 (referring to homologous bovine opsin positions), which is predictive for opsin chromophore binding, as well as the NPXXY motif and the tripeptide motif on positon 310–312 crucial for G-protein activation (*Figure 1D*). The latter matches the NKQ motif, which is characteristic for c-opsins but is also present in some sequences of the recently described group of xenopsins (*Ramirez et al., 2016*) and a few cnidops. Correct placement of the new *L. asellus* opsin is highly relevant for our study and, according to *Ramirez et al. (2016)*, several sequences initially described as c-opsins fall into xenopsins. We therefore performed a thorough phylogenetic analysis with a main focus on the relationship of c-opsins and xenopsins. For this purpose, we ran both maximum-likelihood and Bayesian analyses to check for congruence of different tree inference algorithms, we chose closely related representatives from amongst rhodopsin-like G-Protein coupled receptors as an outgroup, and we extensively tested standard and dataset specific (DS-GTR) evolutionary models for appropriateness.

In order to increase the taxon sampling of potential c-opsins and xenopsins in protostomes, we mined publicly available genomic and transcriptomic resources and *de novo* assembled transcriptomes from publicly available raw Illumina RNA-seq data from 9 nemerteans, 10 flatworms, 7 aceoelomorphs and *Xenoturbella* and from new RNA-seq data from the annelid *Owenia fusiformis*. An initial maximum-likelihood run with RAxML yielded low support for many basal branches. Thus, we performed a leaf-stability analysis based on the bootstrap results to identify rogue sequences. One sequence of *Strongylocentrotus purpuratus* (*opsin2*) and one of the brachiopiod *Lingula anatina* (*peropsin A*) as well as ctenopsin sequences exhibited leaf-stability values less than 0.55. As we were interested in whether ctenopsins are closely allied to c-opsins and xenopsins, only the *opsin2* and *peropsin A* sequences from *S. purpuratus* and *L. anatina* were removed from the dataset. Subsequent maximum-likelihood and Bayesian runs yielded nearly identical tree topologies. In line with the study by *Ramirez et al. (2016)*, c-opsins and xenopsins form distinct and highly supported groups (*Figure 2*, *Figure 2—figure supplement 1*). The new opsin from *L. asellus* groups with high support within xenopsins and is thus termed *L. asellus* xenopsin. In addition to the xenopsins reported by *Ramirez et al. (2016)* from mollusks, brachiopods and rotifers, we discovered this opsin type in *Owenia fusiformis*, a member of the basally branching annelid group of Oweniidae, in several polyclad flatworms and in the triclad flatworm *Schmidtea mediterranea*. By contrast, no previously unidentified c-opsin sequence showed up in any new major animal group. Other groups might be devoid of both xenopsins and c-opsins. We did not find any c-opsin or xenopsin in the nine screened RNA-seq datasets from acoelomorphs or xenoturbellids. Although reasonable numbers of opsins were found in some of those species, reciprocal blast suggests that these proteins are mainly related to r-opsins. The situation is more ambiguous in nemerteans. From the nine screened RNA-seq datasets, we found higher numbers of opsin sequences only in *Lineus longissimus*, but no hints

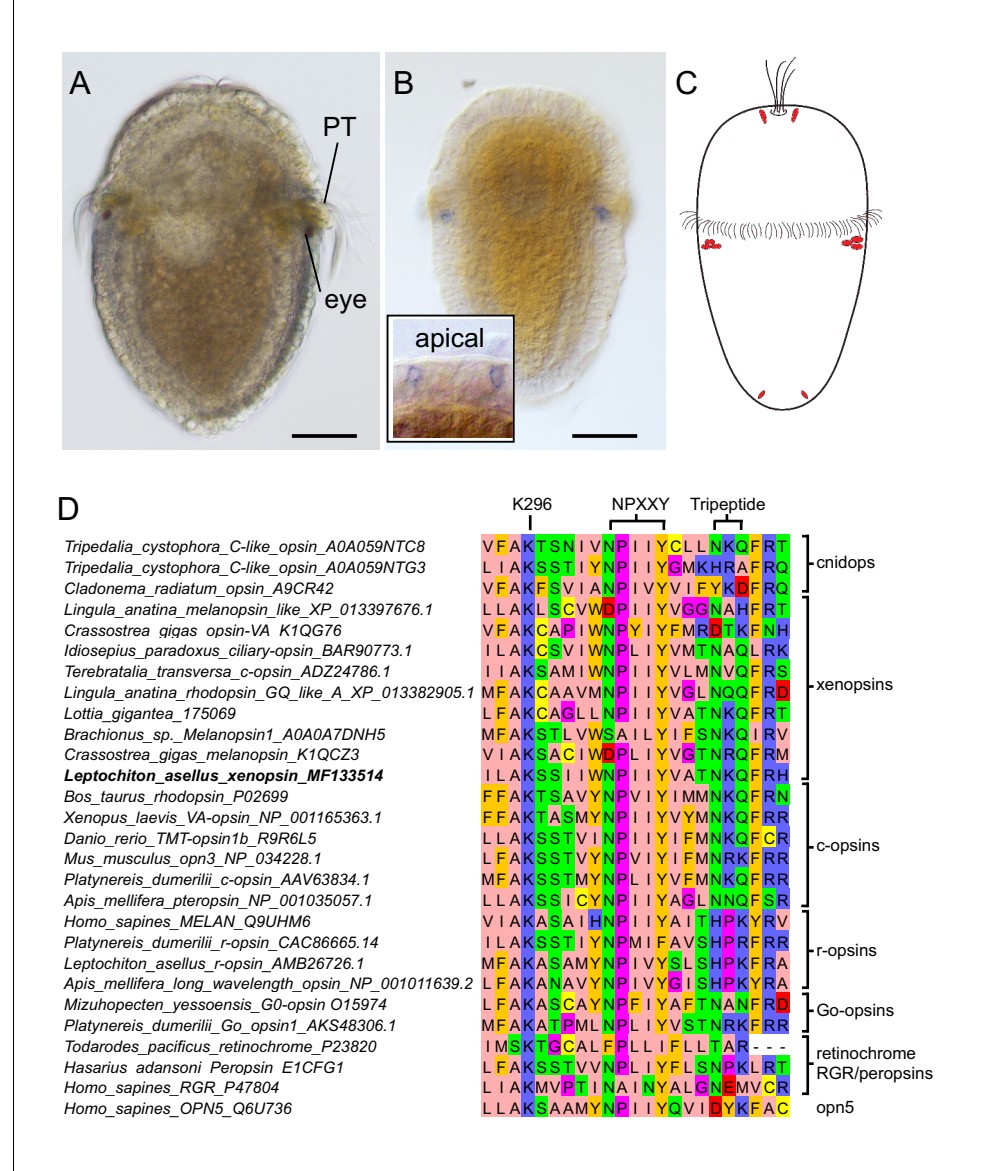

**Figure 1.** Expression of a new visual pigment (xenopsin) in the *L. asellus* larva. (a) In 7 dpf larvae, the eyespot can clearly be seen in a position posterior to the prototroch (PT). (b) The new *L. asellus* opsin is expressed in the area of the larval eyes, as well as in cells in the apical area and at the very posterior end of the larva. The opsin+ cells at the posterior end can only be seen by confocal imaging because of the small cell size. (c) Schematic drawing of the distribution of the cells expressing the new *L. asellus* opsin in the larva. (d) The xenopsin found in *L. asellus* resembles the well-characterized group of c-opsins in containing the tripeptide NKQ, which is important for opsin and G-protein interaction. In contrast to this, the earlier described *Las-r-opsin* exhibits an HPK motif, which is specific for the r-opsin group. (Scalebars 100 μm in (a) and (b).)

DOI: https://doi.org/10.7554/eLife.23435.003

The following figure supplement is available for figure 1:

**Figure supplement 1.** Alignment region showing Lys 296, NPXXY and the tripeptide motif for all opsin sequences included in the study.

DOI: https://doi.org/10.7554/eLife.23435.004

that there are c-opsins or xenopsins in any nemertean species. The low yield of opsin sequences may reflect the real situation or may be due to low sequencing coverage or the tissue samples taken.

Further, our data suggest that xenopsins diversified at an early stage of protostome evolution. Xenopsin paralogs of several platyhelminth species are distributed across two well-supported

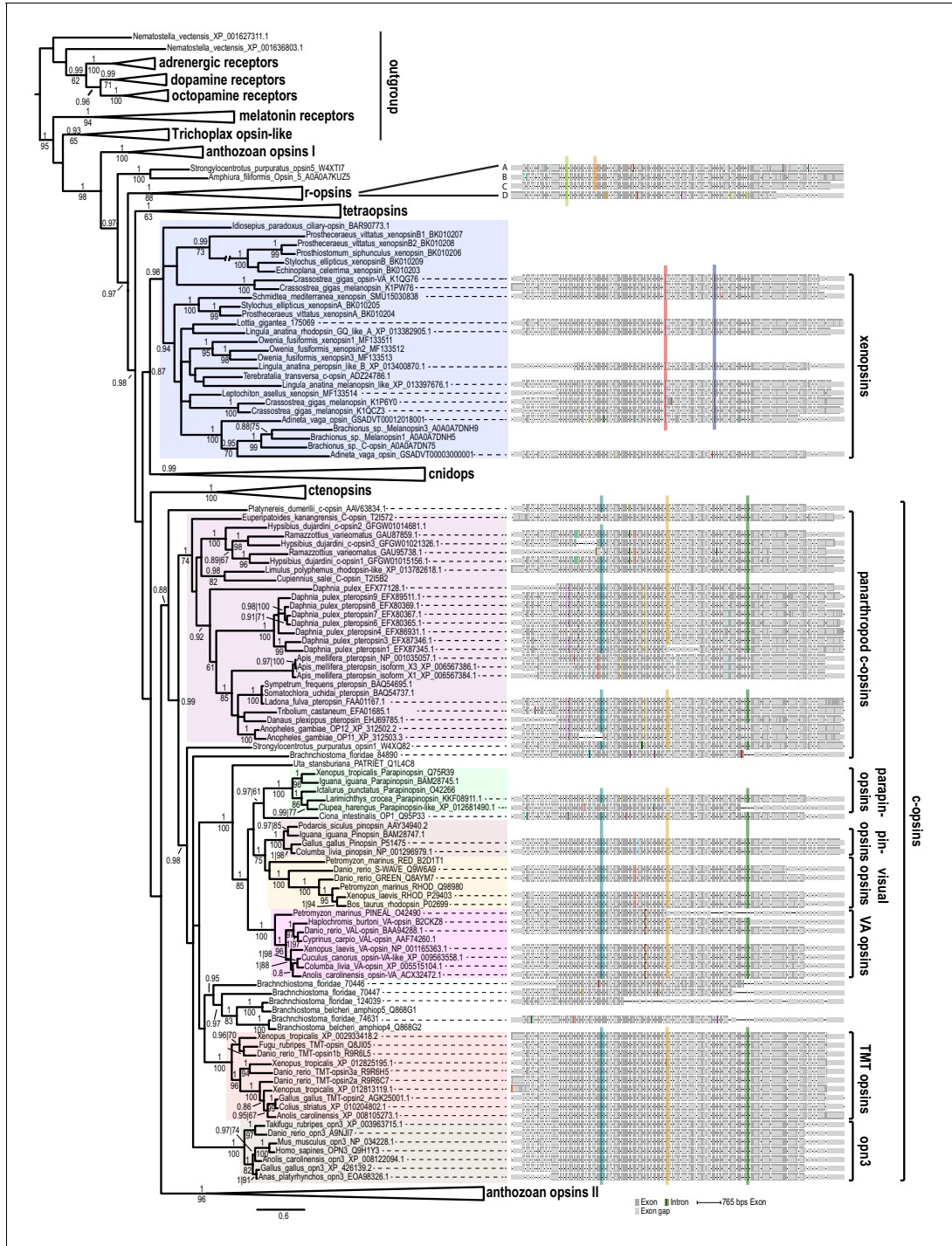

**Figure 2.** Opsin phylogeny and gene structure analysis. Majority consensus tree of Bayesian analysis (Phylobayes, DS-GTR + Γ, 60,000 cycles). Bayesian posterior probabilities and bootstrap values from parallel RAxML analysis are shown. See *Figure 2—figure supplement 1* for uncollapsed tree and *Figure 2—source data 1* for gene accession numbers. Intron positions (colored bars) are mapped on the uncurated protein sequence alignment and conserved positions are highlighted by bars spanning several sequences. See *Figure 2—figure supplement 2* for full opsin gene structures and *Figure 2—figure supplement 3* for intron phases. The new *L. asellus* opsin falls with high support into the group of xenopsins, which is only distantly related to c-opsins. All major c-opsin subgroups share three conserved intron positions. While cnidops that form the sister group of xenopsins lack introns, the two intron positions that are conserved in xenopsins do not overlap with those of c-opsin introns, indicating independent evolutionary origins for xenopsins and c-opsins. The r-opsin sequences included in the gene structure representations are those of A: *Apis mellifera* UV opsin; B: *Limulus polyphemus* UV-like opsin; C: *Crassostrea gigas* r-opsin 10013541; and D: *Homo sapiens* melanopsin.

DOI: https://doi.org/10.7554/eLife.23435.005

*Figure 2 continued on next page*

*Figure 2 continued*

The following source data and figure supplements are available for figure 2:

**Source data 1.** Accession numbers of sequences used for opsin tree inference.
DOI: https://doi.org/10.7554/eLife.23435.009
**Figure supplement 1.** Uncollapsed tree of Bayesian phylogenetic opsin analysis (Phylobayes, dataset specific model DS-GTR + Γ, 60,000 cycles).
DOI: https://doi.org/10.7554/eLife.23435.006
**Figure supplement 2.** Opsin gene structures mapped onto a protein alignment of all opsins included in the phylogenetic analysis, where genomic information was publicly available or generated by this study as for *P. dumerilii* c-opsin and *L. asellus* xenopsin.
DOI: https://doi.org/10.7554/eLife.23435.007
**Figure supplement 3.** Intron phase analysis of all opsins included into the phylogenetic analysis, where genomic information was publicly available or generated by this study, showing that the type-specific introns in c-opsins and xenopsins are conserved not only by position but also by intron phase.
DOI: https://doi.org/10.7554/eLife.23435.008

xenopsin subgroups, one of which also contains sequences from many other protostome taxa (*Figure 2*, *Figure 2—figure supplement 1*). Notably, the tripeptide motif that is crucial for G-protein activation shows a pattern similar to that found in c-opsins (NKQ) within that xenopsin subgroup, which has a broad taxonomic distribution, and in the sequence of *Idiosepius paradoxus*, while this pattern differs considerably in other xenopsins (*Figure 1—figure supplement 1*).

## Gene structure supports independent origins of xenopsins and c-opsins irrespective of their mutually exclusive distribution

*Ramirez et al. (2016)* reported that xenopsins were secondarily lost in several major animal groups. Our data support this view and the same is obviously true for c-opsins. Thus, we specifically compared the presence of c-opsins and xenopsins across bilaterian animals. Even though genomic resources were screened for many species, stunningly we could not recover a single species possessing both c-opsins and xenopsins. This also holds true for higher taxonomic levels, since as sequences of just one of the two opsin groups were found in deuterostomes and in major protostome clades such as mollusks, rotifers, brachiopods, flatworms, arthropods and tardigrades. As the only exception, we found xenopsins (but no c-opsins) in the annelid *Owenia fusiformis*, whereas other annelids such as *Platynereis dumerilii* possess only c-opsins or lack both types as do *Capitella teleta* or *Helobdella robusta* (*Figure 2*, *Figure 2—figure supplement 1*). Such a mutually exclusive taxonomic distribution of protein subfamilies is difficult to explain from an evolutionary perspective and raises the question of whether the distinction of xenopsins and c-opsins may be caused by tree inference artifacts. Thus we performed gene structure analyses in order to obtain independent information on this matter.

Position and phase of introns were determined for all those sequences for which genomic information was publicly available and the new sequences for *L. asellus* opsin and *P. dumerilii* c-opsin, for which the whole gene sequences cloned from genomic DNA. The obtained data were than mapped onto the un-curated protein alignment to identify introns located in homologous sequence positions (*Figure 2*, *Figure 2—figure supplement 2*). This revealed highly conserved and specific exon–intron patterns for many major opsin groups. The patterns differ considerably between xenopsins and c-opsins. Three introns are highly conserved in position and phase (*Figure 2*, *Figure 2—figure supplement 2*) and phase (*Figure 2—figure supplement 3*) throughout protostome c-opsins, deuterostome encephalopsins, TMT opsins, VA-opsins, parapinopsins and pinopsins, and the vertebrate visual opsins. By contrast, all xenopsins share two different intron positions with conserved intron phase (*Figure 2—figure supplements 2 and 3*), with the exception of one sequence from *Adineta vaga*. While the second xenopsin intron position is clearly distinct from those of any c-opsin intron, the first xenopsin intron has a position similar to that of the second c-opsin intron. In order to investigate whether the minor positional difference between these two c-opsin and xenopsin introns are potentially caused by alignment artifacts rather than independent evolutionary origin, we mapped the gene structures onto several different alignments obtained with MAFFT- E-INS-i, MAFFT-G-INS-I, GLProbs, ProbCons and MUSCLE. In all cases, the three conserved c-opsin and two xenopsin introns are recovered. The position of the second c-opsin intron is not stable only in the MUSCLE alignment. This coincides with the poor quality of the MUSCLE alignment in this region

when compared to the other alignment variants. The xenopsin and c-opsin intron positions do not overlap in any of the alignments.

Thus, gene structure provides strong evidence that xenopsins and c-opsins are indeed distinct opsin subgroups, irrespective of their mutually exclusive distribution amongst bilaterian animals. Due to the lack of introns in cnidops, the gene structure analyses cannot provide further support for the sistergroup relationship of cnidops and xenopsins. Likewise, ctenopsins show a very characteristic gene structure and subdivision into two groups, but no clear similarities to c-opsins or xenopsins exist. Notably, *Strongylocentrotus purpuratus* opsin2 and *Lingula anatina* peropsin-like A, which were excluded from the current analysis due to low leaf stability, share conserved introns (*Figure 2—figure supplement 2*), strongly corroborating the very small but ancient group of bathyopsins reported by *Ramirez et al. (2016)*.

## Las-xenopsin is exclusively expressed in r-opsin + PRCs

In order to investigate the expression pattern of *Las-xenopsin*, we performed in-situ hybridization analyses in different larval stages of *L. asellus*. This revealed expression in the posttrochal eye region (*Figures 1B* and *3F*) as well as in cells at the very apical (*Figure 1B* inset) and posterior ends of the larva (*Figure 1C*), an expression pattern strongly resembling that of *Las-ropsin* (*Vöcking et al., 2015*). Double in-situ hybridization unambiguously showed cellular coexpression of *Las-xenopsin* and *Las-ropsin* in all PRCs (*Figures 3F* and *4A*).

## Ciliary features of the otherwise rhabdomeric PRCs

The eye PRCs of *L. asellus* have been shown to be largely of rhabdomeric organization and the r-opsin protein is most likely localized in the microvillar extensions (*Vöcking et al., 2015*). The only xenopsin with known expression site is from the brachiopod *T. transversa*. It was originally described as a c-opsin and is expressed in the ciliary PRCs of the larval eyes. For this reason, we searched for indicators of cilia development in the PRCs of *L. asellus* by means of in-situ hybridization, i.e. we looked for the transcription factors *Foxj1* and *RFX*, which are involved in ciliary development in

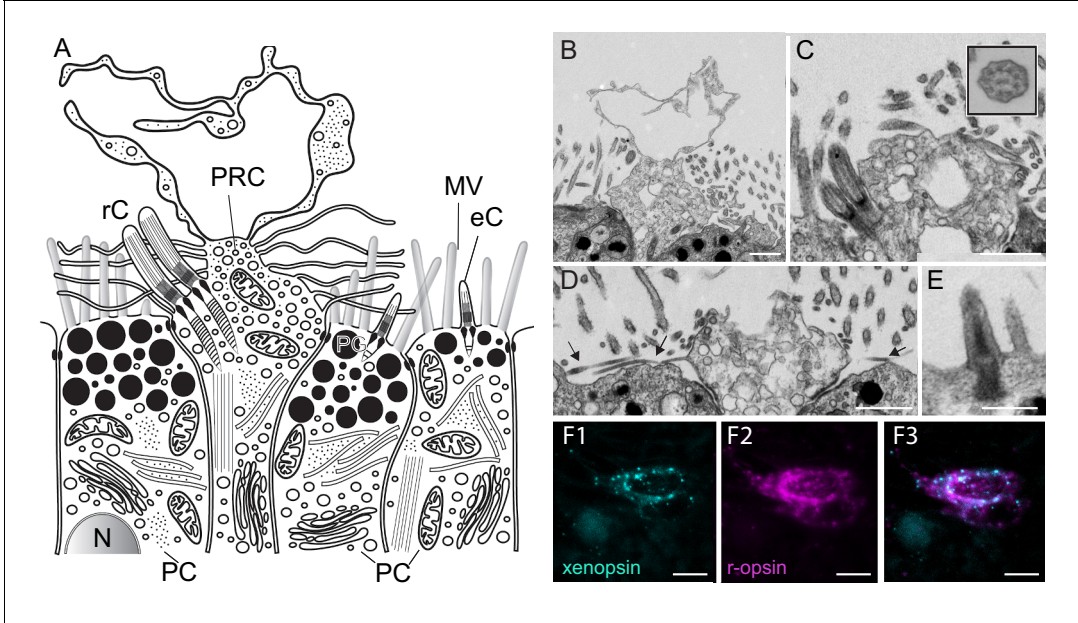

**Figure 3.** Fine structure of the eye photoreceptor cell surface and co-expression of *Las-xenopsin* and *Las-ropsin* in the eye. (a) Schematic representation of eye structure based on serial section EM data. (b–d) EM micrographs of the apical surface of eye photoreceptor cells (PRCs). (b) Microvillar-like vertical extensions of the eye PRC. (c) Two cilia emerging from eye PRC (inset: cross section of eye PRC ciliary axonem showing 9 × 2 + 2 pattern). (d) Horizontal microvilli (*arrows*) emerging from eye PRC. (e) Rudimentary cilium of adjacent epidermal cell. (f1-3) Double in-situ hybridization shows that *Las-xenopsin* and *Las-r-opsin* are coexpressed in eye PRCs. (Scalebars: 2 µm in **b–e**, 5 µm in **f**). eC, epidermis cell cilium; MV, microvilli; N, Nucleus; PC, pigment cell; PG, pigment granules; rC, photoreceptor cell cilium.
DOI: https://doi.org/10.7554/eLife.23435.010

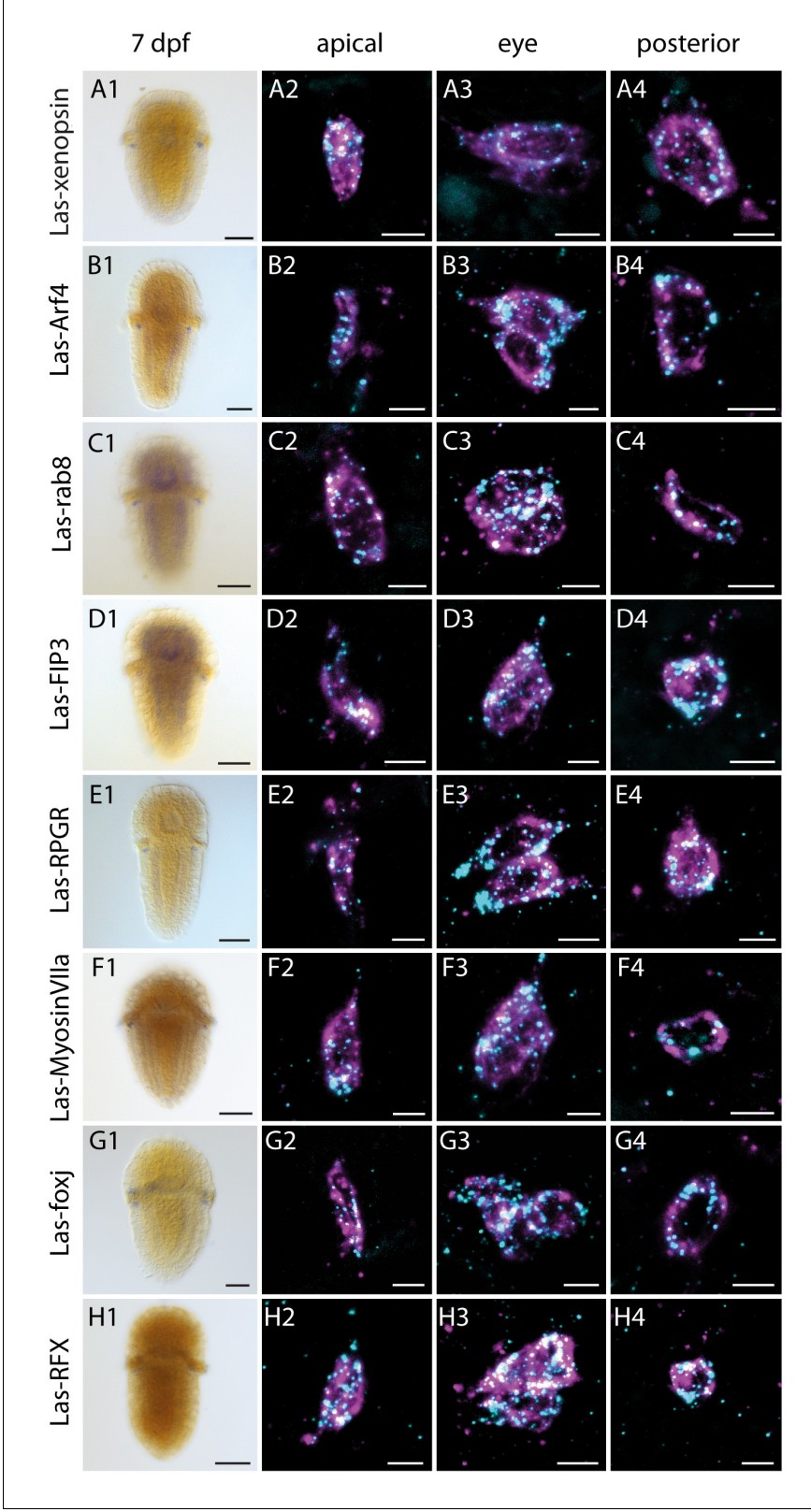

**Figure 4.** Expression of *Las-xenopsin* as well as genes involved in ciliary opsin transport and development of cilia. Column 1: single labeling of gene X. Column 2–4: double labeling of gene X (*cyan*) and *Las-r-opsin* (*magenta*) in the anterior, posttrochal eye and posterior regions. *Las-xenopsin* (**a1–a4**), *Las-Arf4* (**b1–b4**), *Las-rab8* (**c1–c4**), *Las-FIP3* (**d1-4**), *Las-RPGR* (**e1-4**), *Las-Myosin* VIIa (**f1-4**), *Las-foxj1* (**g1-4**) and *Las-RFX* (**h1-4**) are coexpressed with *Las-r-*

*Figure 4 continued on next page*

*Figure 4 continued*

*opsin* in anterior, eye and posterior PRCs. Additionally, *Las-rab8* (c1) and *Las-FIP3* (d1) show a broader expression in the nervous system of 7 dpf larvae and *Las-foxj1* shows expression also in the apical area and prototroch cells of young larvae (*Figure 4—figure supplement 1*). See *Figure 4—figure supplements 1–3* for gene trees of Arf4, FIP and RFX. (Scalebars: 100 µm in column **1**; 5 µm in columns **2–4**).
DOI: https://doi.org/10.7554/eLife.23435.011

The following source data and figure supplements are available for figure 4:

**Figure supplement 1.** *Las-foxj1* expression in a young larva (48 hpf).
DOI: https://doi.org/10.7554/eLife.23435.012

**Figure supplement 2.** Alignment of C-termimal ends of the opsins involved in this study.
DOI: https://doi.org/10.7554/eLife.23435.013

**Figure supplement 3.** Arf4 evolution.
DOI: https://doi.org/10.7554/eLife.23435.014

**Figure supplement 3—source data 1.** Accession numbers of sequences used for Arf4 tree inference.
DOI: https://doi.org/10.7554/eLife.23435.017

**Figure supplement 4.** FIP evolution.
DOI: https://doi.org/10.7554/eLife.23435.015

**Figure supplement 4—source data 1.** Accession numbers of sequences used for FIP tree inference.
DOI: https://doi.org/10.7554/eLife.23435.018

**Figure supplement 5.** RFX evolution.
DOI: https://doi.org/10.7554/eLife.23435.016

**Figure supplement 5—source data 1.** Accession numbers of sequences used for RFX tree inference.
DOI: https://doi.org/10.7554/eLife.23435.019

different animals (*Alten et al., 2012*; *Choksi et al., 2014*; *Dubruille et al., 2002*; *Yu et al., 2008*). Both are expressed in the PRCs of 7 dpf *L. asellus* larvae (*Figure 4G,H*, *Figure 4—figure supplement 1*) and *Las-Foxj1* was also found to be expressed in the prototroch cells and the apical organ of younger larvae (*Figure 4—figure supplement 1*). As this points towards the formation of cilia by the PRCs, we screened the apical surface of the PRCs for cilia by means of electron microscopy (*Figure 3A–E*). Indeed, we found two cilia in each PRC in addition to the many microvillar extensions (*Figure 3A,C,D*), corroborating ultrastructural data from other chitons (*Bartolomaeus, 1992*; *Fischer, 1980*; *Rosen et al., 1979*). The cilia show a $9 \times 2 + 2$ microtubule pattern (*Figure 3C inset*). Neither the cilia nor the microvillae exceed the cuticle nor exhibit specialized surface extensions (*Figure 3A,C*), but they are longer than the rudimentary cilia found in surrounding epithelial cells and do not taper (*Figure 3A,C,E*). As we were interested in whether the cells feature a mechanism to transport opsin into cilia, we looked into transporters that are known to be involved in ciliary opsin targeting. Although the C-terminal VxPx motif known for cargo binding in vertebrate (but not protostome) c-opsin is not conserved in xenopsins (*Figure 4—figure supplement 2*), several ortholgos of the general players of ciliary targeting that are also active in ciliary opsin targeting, namely *Las-Arf4*, *Las-rab8*, *Las-FIP3*, *Las-RPGR* and *Las-MyosinVIIa* (*Liu et al., 1999*; *Wang and Deretic, 2014*, *2015*; *Wang et al., 2012*), were found to be specifically expressed in the PRCs by double in-situ hybridization with *Las-r-opsin* (*Figure 4B–F*, *Figure 4—figure supplements 3–5*).

## Discussion

In vertebrates, c-opsins drive the important process of vision, as they constitute the visual pigments of the retinal rods and cones and they serve additional functions when expressed in the pineal or in deep brain PRCs. Thus, the identification of c-opsins in protostome invertebrates provided the possibility to explore the role of c-opsin+ PRCs in distant relatives and to uncover the evolutionary roots of vertebrate vision. For a long time, only very few protostome c-opsins were known and these unambiguously group within the clade of c-opsins. This narrowed down the choice of potential study subjects. Basically, there were few pteropsins from arthropods and one sequence from the annelid *Platynereis dumerilii* (*Arendt et al., 2004*; *Eriksson et al., 2013*; *Velarde et al., 2005*).

In addition, new protostome sequences, which resemble c-opsins in certain aspects, were not instantly included in broad opsin analyses and this led to weakly supported orthology assignments

that suggested a broader distribution of c-opsins in protostome invertebrates. Seemingly, this is not the case. Initial evidence that the new sequences do not encode c-opsins, but are closer affiliated to cnidops, has been provided by *Hering and Mayer (2014)*, though with low support. This new group included sequences of hitherto uncharacterized visual pigments from the mollusks *Crassostrea gigas* and *Lottia gigantea*, a sequence from the brachiopod *T. transversa* that originally was described as c-opsin (*Passamaneck et al., 2011*) , and a weakly associated sequence from *Strongylocentrotus purpuratus* (*opsin2*) that did not group with other major opsin sequence groups in other analyses (*D'Aniello et al., 2015*; *Ooka et al., 2010*; *Raible et al., 2006*). Probably because of the increased taxon sampling and formal tree reconciliation, the recent phylogenetic study of *Ramirez et al. (2016)* provides for the first time strong support for a new group of xenopsins, which are distinct from c-opsins. We show that irrespective of the remarkable mutually exclusive taxonomic distribution of xenopsins and c-opsins, this distinction does not rely on tree inference artifacts, as can be seen from the specific gene structures of both opsin types.

Our data reveal a broader occurrence of xenopsins in protostomes than hitherto anticipated, pointing towards a high relevance of this opsin type in animal physiology. Accordingly, xenopsins are present not only in mollusks, rotifers and brachiopods as already reported by *Ramirez et al. (2016)*, but also in several flatworms and in Oweniidae, the first branch within annelids. The situation in nemerteans is not clear because the screened RNA-seq datasets are probably not representative due to the low total number of opsins found. In accordance with *Ramirez et al. (2016)*, we have no evidence that xenopsins exist in deuterostomes. *Strongylocentrotus purpuratus opsin2* is the only deuterostome sequence that has ever been suggested to show affinities to sequences now classified as xenopsins (*Hering and Mayer, 2014*). However, the high similarity of the gene structure to that of *Lingula anatina* peropsin A strongly supports a position of both sequences in the ancient small group of bathyopsins sensu *Ramirez et al. (2016)*, which exclusively comprises only a few echino-derm and brachiopod sequences.

Nevertheless, the data provide clear evidence that xenopsins were already present in the last common ancestor of deuterostomes and protostomes. In the analysis of *Ramirez et al. (2016)* and in our study, xenopsins form the sister group to cnidops, and this may very likely reflect the evolu-tionary history. As the basal interrelationships of the major opsin groups were shown to be strongly affected by outgroup composition, taxon sampling, tree inference algorithms and parameters (com-pare *Cronin and Porter, 2014*; *Feuda et al., 2012*, *2014*; *Hering and Mayer, 2014*; *Porter et al., 2012*), this hypothesis may, however, need further investigation. Unfortunately, gene structure data are not informative due to the lack of introns in cnidops. For this reason and in contrast to *Ramirez et al. (2016)*, we apply the term xenopsin only to bilaterian sequences and not to cnidops. Nevertheless, the assumption that xenopsins emerged within protostomes would require a sister group relationship between xenopsins and another clade of protostome-specific opsins. This would need major rearrangements of opsin tree topology and is not supported by gene structure data. In the (unlikely) scenario that xenopsins do not turn out to be sister to cnidops, it remains to be deter-mined whether xenopsins were already present in the ancestor of Bilateria or emerged in the last common ancestor of protostomes and deuterostomes. This depends on the final phylogenetic posi-tion of acoelomorphs in relation to Bilateria and on further upcoming sequence data. We could not detect any xenopsin or c-opsin in the screened RNA-seq data from seven acoelomorphs and one xenoturbellid.

According to our data, frequent lineage-specific loss of either xenopsins or c-opsins (or both) dur-ing animal evolution has to be assumed. To date, no explanation can be given as to why the two opsin groups do not co-occur in a single organism. The intriguing possibility remains that an underly-ing causal relation drove the evolution of a large protein family that way. This will be an interesting case to study on the molecular physiological and the genomic levels.

The observed co-expression of xenopsin and r-opsin in the eye PRCs of *L. asellus* is remarkable from a functional perspective. As a general observation, the vast majority of PRCs express only one type of visual opsin. In nearly all cases where more than one visual opsin was found in PRCs, the respective opsins belong to the same opsin subgroup and evoke the same phototransduction cas-cade. This has been shown for r-opsins in several arthropods, as for example in the eye PRCs of *Lim-ulus polyphemus* (*Katti et al., 2010*), fiddler crabs (*Rajkumar et al., 2010*) and a butterfly (*Arikawa et al., 2003*), and for c-opsins in for example mice (*Applebury et al., 2000*), guinea pigs (*Parry and Bowmaker, 2016*), salamander (*Isayama et al., 2014*) and cichlid fishes (*Dalton et al.,*

*2015*). In most cases, expansion or tuning of the visual spectrum is the suggested function. Co-expression of opsins belonging to different, evolutionary distinct opsin types is reported only very rarely, as in the case of the annelid *P. dumerilii,* where Go-opsins and r-opsin are employed in eye PRCs (*Gühmann et al., 2015*). Here, the main function likewise seems to be the expansion of the visual spectrum of the cells. Our findings in *L. asellus* point towards another direction. The *L. asellus* eye PRCs express all relevant elements of standard r-opsin-mediated GNAQ-dependent IP3 signaling (*Vöcking et al., 2015*). By contrast, the NKQ tripeptide motif present in *L. asellus* xenopsin is known to be involved in the GNAI- and cGMP-based signaling (*Marin et al., 2000*) of c-opsins and is similar to motifs in cnidops, some of which were shown to mediate GNAS- and cAMP-based signaling (*Koyanagi et al., 2008*; *Liegertová et al., 2015*). The eye PRCs might thus be able to integrate different light signals directly. To our knowledge, the only case known in which co-expressed opsins evoke competing signaling cascades is in the pineal of the lizard *Uta stasburiana* (*Su et al., 2006*). Functional analysis of the molecular physiology of the *L. asellus* eye PRCs may thus be of great interest to sensory biologists.

Xenopsins are seemingly able to enter cilia. This type of opsin is employed in the purely ciliary PRCs of the larval eyes of the brachiopod *Terebratalia transversa* (*Passamaneck et al., 2011*). We find it expressed in eye PRCs in *L. asellus*, whose microvilli employ r-opsin (*Vöcking et al., 2015*) but which also have cilia. The expression of Arf4, rab8, FIP3 and RPGR provides the possibility that these proteins are involved in xenopsin trafficking, as they are in c-opsin ciliary trafficking in vertebrate rods and cones (*Liu et al., 1999*; *Wang and Deretic, 2014*, *2015*; *Wang et al., 2012*). As the C-terminal VxPx motif necessary for the Arf4-mediated transport of vertebrate c-opsins (*Wang and Deretic, 2014*) is missing in xenopsins, Arf4-mediated transport of xenopsin would require other protein interactions. Interestingly, the same holds true for ciliary transport of cnidops and of c-opsins from organisms other than vertebrates and tunicates, as the VxPx motif is only conserved in the latter opsin groups and not in cephalochordate, echinoderm or protostome c-opsins or cnidops, some of which have been shown to be expressed in ciliary PRCs (*Arendt et al., 2004*; *Bielecki et al., 2014*; *Kozmik et al., 2008*; *Vopalensky et al., 2012*). It may be that Arf4 binding to c-opsins other than those of vertebrates and tunicates, xenopsins and cnidops relies on other protein interactions as in the case of vertebrate c-opsins. As an alternative, Arf4-complex-mediated ciliary opsin trafficking has to be regarded as a chordate invention.

Our extensive data mining did not change the picture that arthropods and certain annelids are the only organisms in which insights can be obtained on the physiological role of the protostome counterparts of vertebrate visual pigments. C-opsins have been found in no other protostomes. As outlined above, xenopsin-employing PRCs are promising study subjects for many other questions in sensory biology. Furthermore, we propose that data on the employment of xenopsins have significant impact on the understanding of the evolution of animal eyes and of photoreceptor cell types in general. As discussed below, we regard the presence of xenopsins in ancestral eyes as a likely scenario. This implies that the occurrence of xenopsins in the eyes of protostomes is not due to the flexible recruitment of opsins in existing eyes during evolution or to frequent invasion of new kinds of photoreceptor cells into eyes, but rather to inherited employment in an ancestral eye photoreceptor cell type.

The wide distribution of r-opsin+ cells in the eyes of protostomes and deuterostomes, alongside the reported similarities in molecular physiology and the process of cell specification, have been interpreted by many studies as a sign of common evolutionary origin and the presence of r-opsin + rhabdomeric PRCs in the eye of the last common ancestor of bilaterians (*Arendt, 2008*; *Arendt et al., 2002a*; *Koyanagi et al., 2005*; *Lamb, 2013*; *del Pilar Gomez et al., 2009*). Data on c-opsins, however, were interpreted in different ways. The pivotal role that c-opsins play in the rods and cones of vertebrate eyes is obvious and findings from the cephalochordate frontal eye suggest that c-opsin+ ciliary PRCs were the predominant eye sensory cells already in the last common ancestor of chordates (*Vopalensky et al., 2012*). In protostomes, c-opsins were initially described only in the brain (*Arendt et al., 2004*; *Beckmann et al., 2015*; *Velarde et al., 2005*), which corresponds to their broad occurrence in the brain of basally branching deuterostome c-opsins such as encephalopsins (*Blackshaw and Snyder, 1999*; *Nissilä et al., 2012*). Thus, it has been suggested that c-opsin + PRCs did originally not form part of eyes, but were rather localized deeply in the brain and not associated with screening pigment (*Arendt, 2008*; *Arendt et al., 2004*; *Lamb, 2013*). They secondarily invaded the eyes of chordates, but stayed within the brain in protostomes.

The presence of ciliary PRCs revealed by structural studies of the eyes of some protostome groups (*Purschke et al., 2006*; *Salvini-Plawen, 2008*; *Woollacott and Eakin, 1973*; *Woollacott and Zimmer, 1972*) would contradict this scenario if those cells showedmolecular characteristics similar to those of c-opsin+ brain PRCs. Accordingly, the discovery of a c-opsin that is expressed in ciliary eye PRCs of a brachiopod put forward alternative explanations favoring a higher lability of PRC function and multiple independent recruitments of r-opsin+ and c-opsin+ PRCs for directional vision in animal eye evolution (*Passamaneck et al., 2011*). These data conflicts are resolved by the fact that all c-opsins that were previously reported from the visual cells of protostome eyes turned out to be xenopsins instead of c-opsins, according to *Ramirez et al. (2016)* and our study . The only exception relies on a PCR-based report of a true c-opsin in an onychophoran eye (*Eriksson et al., 2013*), but this is contradicted by a cellular expression analysis by *Beckmann et al. (2015)* showing that c-opsin is absent from the eye retina layer and only present deeply within the brain and the optical ganglia.

By contrast, all existing data on the expression of xenopsins point towards their presence in eyes. The opsin reported from brachiopod larval eyes (*Passamaneck et al., 2011*) is a xenopsin. We detected xenopsin in eye PRCs of a mollusk and in a few extraocular PRCs, which are regarded as derivatives of cerebral eyes (*Vöcking et al., 2015*). RNA-seq data from cephalopods likewise suggest the presence of xenopsins in eyes (*Yoshida et al., 2015*). One option to explain the appearance of xenopsin in protostome eyes is to assume repeated recruitment of xenopsins into this context. This implies the co-option of xenopsin by ancestral microvillar r-opsin+ eye PRCs in eyes like those of *L. asellus*. In purely ciliary eyes such as those in brachiopods, either the formerly r-opsin+ cells completely changed identity or the original PRCs were replaced by new ciliary xenopsin+ PRCs, or the whole eyes arose completely *de novo* while the ancestral microvillar eyes were reduced. Abrupt changes and new inventions would be important and frequent events in eye evolution.

Alternatively, assuming the presence of both r-opsin and xenopsin in ancestral eyes provides probably a simpler explanation (*Figure 5*). Notably, this is not in conflict with the absence of xenopsins in the eyes of well-studied protostomes such as arthropods, or certain annelids, as secondary loss of xenopsin in the respective lineages is clearly evident from sequence resources and opsin phylogeny. R-opsin and xenopsin may initially have been employed in different cells, but a very attractive hypothesis is that cellular co-expression of r-opsin and xenopsin, like that which we observed in the eyes of *L. asellus*, is not an exceptional case but is inherited from ancestral eye PRCs that employed both opsins and exhibited microvilli and cilia (*Figure 5*). The cerebral eye PRCs of mixed microvillar/ciliary organization described in some organisms (*Blumer, 1996*; *Hughes, 1970*; *Zhukov et al., 2006*) may have barely maintained the ancestral organization, whereas microvillar eye PRCs may have emerged by reducing the cilia in taxa in which xenopsin was secondarily lost and the only visual pigment that survived is that associated with microvilli. Vice versa, purely ciliary eye PRCs, as have been described in several protostomes (*Blumer, 1994*, *1999*; *Passamaneck et al., 2011*; *Woollacott and Eakin, 1973*; *Zimmer and Woollacott, 1993*), may have formed coincidently with downregulation or complete loss of r-opsins. The ancestral r-opsin+ and xenopsin+ PRC type may have arisen within the protostome branch leading to lophotrochozoan organisms (variant A in *Figure 5*), but even a much earlier origin is conceivable (variant B in *Figure 5*) as the absence of xenopsins in deuterostome eyes can easily be explained by the obvious secondary loss of this visual pigment in these animals.

The scenario outlined above links back to older debates about PRC evolution and the ancestral function and significance of vestigial cilia found in many invertebrate microvillar eye PRCs. The vestigial ciliary structures described vary in the length and in the structural organization of the microtubular axonem and the ciliary rootlet, and often only basal bodies are found (*Eakin and Westfall, 1964*; *Eakin, 1972*; *Turbeville, 1991*; *Verger-Bocquet, 1992*). These structures were often interpreted as remnants of the motile cilia of the epidermal cells from which the PRCs initially originated (*Arendt et al., 2009*; Eakin *1979*, *1982*; *Mayer, 2006*). Derivation from sensory cilia was only assumed in the context of a hypothesis proposing that animal eye PRCs originated multiple times within the different animal groups from inconspicuous cells enrolled in a diffuse dermal light sense (*von Salvini-Plawen and Mayr, 1977*; *Salvini-Plawen, 2008*). This thesis was contradicted by the upcoming molecular physiological and developmental data favoring conserved evolutionary lineages of r-opsin+microvillar and c-opsin+ ciliary PRC types (*Arendt, 2008*; *Arendt et al., 2004*; *Fernald, 2006*; *Lamb, 2013*; *Shubin et al., 2009*). However, the observed cellular co-expression of

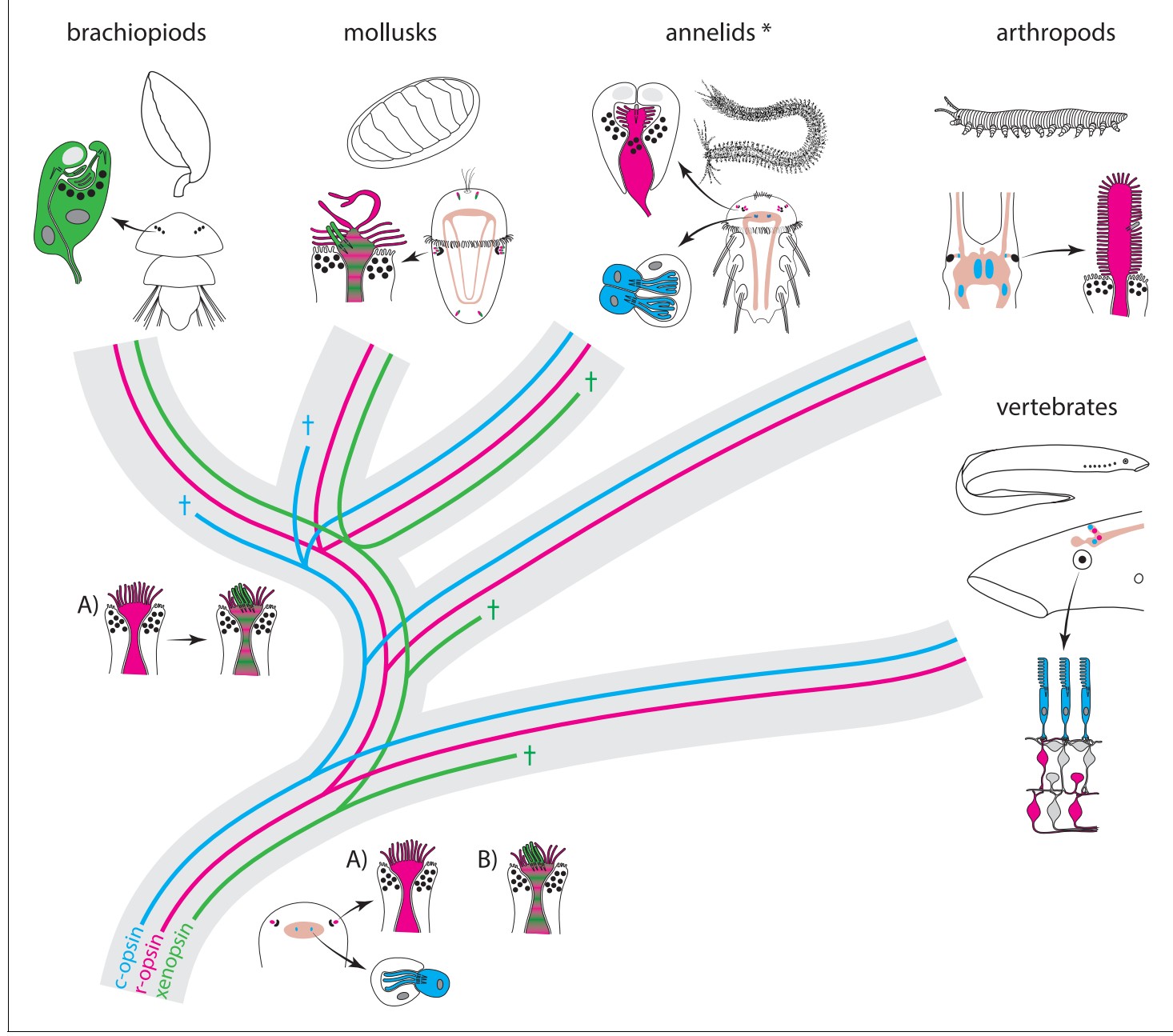

**Figure 5.** Scenarios for eye PRC evolution. C-opsin, xenopsin and r-opsin are present in the bilaterian ancestor. Scenario A. In the bilaterian ancestor, c-opsin is employed by ciliary unpigmented brain PRCs not serving directional vision. Initially the only eye PRCs are of the r-opsin+ microvillar PRC type. In protostomes, c-opsin+ PRCs stayed in the brain and do not form part of the eye retina layer in recent arthropods and annelids, but became integrated into the eye retina of vertebrates. They were reduced in mollusks and brachiopods coincident with gene loss of c-opsin. One option preventing the assumption of multiple co-option of xenopsin by eye PRCs or emergence of completely new PRCs or eyes is that xenopsin was recruited once by r-opsin+ eye PRCs in the stem lineage of lophotrochozoans and entered sensory cilia. Such cells are still present in basal mollusks. In annelids except the basally branching Oweniidae (labelled with asterisk), xenopsin is lost and the eye PRCs are turned into purely microvillar cells. In brachiopod larvae, xenopsin is maintained and present in the purely ciliary eye PRCs. Scenario B. The evolution of ciliary c-opsin+PRCs is the same as in Scenario A. However, already the eyes of the bilaterian ancestor employed a very plastic mixed r-opsin+ and xenopsin+ rhabdomeric/ciliary PRC type. Distribution of PRC types and eye PRC organization in Bilateria is largely the result of lineage-specific loss of either xenopsin or c-opsin. Loss of xenopsin in vertebrates, arthropods and most annelids led to microvillar r-opsin+ PRCs. Though still present in the genome, r-opsin became downregulated in eye PRCs of brachiopods. Schemes are partly based on drawings and images from *Rhode (1992)* and *Passamaneck et al. (2011)*.
DOI: https://doi.org/10.7554/eLife.23435.020

r-opsin and xenopsin in cells bearing microvilli and cilia now suggests that vestigial cilia in microvillar PRCs may indeed also go back to cilia with sensory functions. It is worth noting that the often inconspicuous primary cilia known from most kinds of vertebrate cells are considered to be sensory organelles that are important for proper cell function, intercellular signalling and development (*Malicki and Johnson, 2017*; *Singla, 2006*; *Walz, 2017*). Last but not least, the distinction between motile and sensory cilia is no longer regarded as clear cut. More and more motile cilia that also display sensory capabilities are being reported (*Kleene and Van Houten, 2014*; *Shah et al., 2009*; *Warner et al., 2014*) and the distribution across eukaryotes of molecular components relevant for locomotion and sensation suggests that cilia were enrolled in both functions upon their emergence (*Bloodgood, 2010*; *Johnson and Leroux, 2010*; *Mitchell, 2007*, *2017*; *Quarmby and Leroux, 2010*). The same is conceivable for cilia in ancient photoreceptor cells.

As current data only show use of xenopsin, but not of c-opsin, in the ciliary structures of protostome cerebral eyes, the described scenario is in line with the hypothesis that the cerebral eyes of animals employ only few different PRC types. This scenario does not require the assumption of multiple recruitments of sensory cilia and/or visual pigments or PRCs by cerebral eyes or of multiple origins of those entities. Instead, the PRCs employed in prostostome cerebral eyes are rendered as representatives of a highly plastic cell type. The ratios of microvilli/cilia and of r-opsin/xenopsin, and in consequence the downstream targets and the physiological properties of the PRCs, may have changed repeatedly within evolutionary lineages unless either the r-opsin or the xenopsin genes were finally lost. Subsequent studies on xenopsin+ PRCs may thus strongly impact the understanding of the degree to which conserved cell types and the properties of sensory cells can change over time and of how these changes occur.

In conclusion, opsin phylogenetic analyses combined with gene-structure analysis strongly confirm the existence of an old clade of xenopsins, which is evolutionarily distinct from c-opsins. The physiological role of this opsin type is not yet understood and requires further studies, especially in the interesting case of co-expression with r-opsin. The taxonomic distribution and the available epression data suggest that xenopsins may play a pivotal role in tracing PRC and eye evolution in bilaterian animals. The dichotomous classification of eye PRCs into ciliary c-opsin+ and rhabdomeric r-opsin+ cell types may no longer be appropriate, and the documented presence of xenopsin in eye PRCs may be due to common ancestry. The described evolutionary scenario provides simple explanations for the origin of eye PRCs exhibiting both microvilli and cilia and for how purely ciliary PRCs may have emerged in protostome eyes, and it rises intriguing questions of how much conserved cell types can change over time.

## Materials and methods

### *Leptochtion asellus* culture

Adults of *L. asellus* were collected close to the Norwegian coastline in Bergen during the period from September to December 2015 and kept in the dark at 8°C. They were cultured in groups of female and male animals and fertilized egg balls were collected each morning. Larvae were kept on a light/dark cycle of 12h/12 hr at 18°C.

### Gene cloning and RNA probe generation

The assembly of Illumina (San Diego, California, RRID:SCR_010233) HISeq RNA-seq data of 2–11d old larval material described by *Vöcking et al. (2015)* was used to identify transcripts of interest via bidirectional blast. Whole transcripts and fragments were then amplified by PCR using gene-specific primers and cDNA prepared with Super Script II (Thermo Fisher Scientific, Waltham, Massachusetts) RRID:SCR_008452), subsequently ligated into pgemT-easy vector (Promega, Madison, Wisconsin, RRID:SCR_006724) and cloned into Top10 chemically competent *E. coli* (Thermo Fisher Scientific). Sanger sequencing was used to verify the cloned sequences before DIG- and FITC-labeled sense and antisense probes were generated with T7- and SP6-RNA Polymerases (Roche, Basel, Switzerland, RRID:SCR_001326) or with the Megascript Kit (Thermo Fisher Scientific).

## Gene orthology and phylogenetic analyses

Reciprocal blast yielded unambiguous results for gene orthology assignment of *Las-FoxJ1*, *Las-MyosinVIIa*, *Las-rab8a* and *Las-RPGR*. In all other cases, public databases (GenBank (RRID:SCR_002760), JGI (RRID:SCR_002383), Uniprot (RRID:SCR_002380)) and the *Leptochiton* transcriptome were screened for homologs by text search, blast (RRID:SCR_004870) and HMMER (RRID:SCR_005305) with respective query sequences or domain profiles for subsequent gene tree generation based on maximum likelihood and Bayesian inference. Sequence alignments with MAFFT (RRID:SCR_011811) 7.221 (EINSI option) were manually curated. Best fitting substitution models were selected with Prot-Test (RRID:SCR_014628) 3.4 according to Akaike information, Bayesian information and Decision theory criteria or with the RAXML (RRID:SCR_006086) based ProteinModelSelection script by Alexandros Stamatakis. Maximum likelihood analyses were run with RAXML 8.2.8 on the Cipres Science Gateway (RRID:SCR_008439, *Miller et al., 2010*) with gamma modeling of rate heterogeneity and rapid bootstrapping with automatic stop by majority rule criterion autoMRE. Bayesian analyses were run with Phylobayes (RRID:SCR_006402) 4.1 or Phylobayes-MPI 1.6 using the same substitution models chosen for ML analyses and gamma distribution for rate heterogeneity. Generally, four chains were run and tested for convergence with bcomp command discarding the 20% first cycles as burn-in and used for generating majority-rule consensus trees.

## Opsin tree inference

As orthology assignment of the new *L. asellus* opsin sequence obtained from the RNA-seq data is critical for this study, the general routines for phylogenetic inference described above were further adapted to exclude possible artifacts discussed in recent opsin analyses. To evaluate the relationship to c-opsins, ctenopsins, xenopsins, cnidops and anthozoa II opsins *sensu Hering and Mayer (2014)*, a broad sampling of these opsin types across metazoans was included. For this purpose, publicly available sequence resources (GenBank, Uniprot, Genoscope (RRID:SCR_002172), JGI and *Macrostomum lignano* genome initiative (http://www.macgenome.org)) were screened for homologs by text search, as well as by blast and HMMER searches based on a set of ten query sequences sampled from the above-mentioned opsin groups and the new opsin sequence from *L. asellus*. We kept only those sequences that exhibited the PFAM 7tm_1 domain, contained the residue Lys296 critical for chromophore binding in opsins, and yielded representatives from the above-mentioned opsin groups amongst the best hits after reciprocal blast against GenBank. In order to reduce the computation time for downstream analyses, sequences yielding c-opsins as best reciprocal blast hits were down-sampled in taxa such as vertebrates and arthropods, where high numbers of sequences were obtained.

As the number of sequences retrieved was low for several prostostome taxa, such as annelids, nemerteans, platyhelminths and ecdysozoans others than arthropods, as well as for acoelomorphs, which form a potential sister group to all other bilaterians (*Cannon et al., 2016*), we performed a second round of sampling for these taxa following different strategies. Three complete opsin sequences were obtained from the *Hypsibius dujardini* NCBI TSA archive, matching the published incomplete sequences for c-opsin1, c-opsin2, and c-opsin3, which lacked position 296 and thus were initially removed from the dataset. Illumina RNA-seq raw data were downloaded with fastq-dump from the NCBI SRA (RRID:SCR_004891) read archive for the nemerteans *Baseodiscus unicolor, Cerebratulus* sp., *Lineus longissimus, Nipponemertes* sp., *Paranemertes peregrina, Malacobdella grossa, Cephalothrix hongkongiensis, Hubrechtella ijimai, Tubulanus polymorphus,* the flatworms *Stenostomum sthenum, Kronborgia amphipodicola, Geocentrophora applanata, Prorhynchus* sp., *Echinoplana celerrima, Prostheceraeus vittatus, Prosthiostomum siphunculus, Stylochus ellipticus, Itaspiella helgolandica, Microdalylellia schmidti,* the acoelomorphs *Ascoparia* sp., *Diopisthoporus longitubus, Eumecyonostomum macrobursalium, Isodiametra pulchra, Meara stichopi, Nemertoderma westbladi, Sterreria* sp. and the xenoturbellid *Xenoturbella bocki* . From these data and from our own RNA-seq data from several developmental stages from the annelid *Owenia fusiformis*, transcriptome assemblies were generated using Trimmomatic (RRID:SCR_011848) 0.36 for adapter and quality trimming, Rcorrector (*Song and Florea, 2015*) for read error correction and trinity-2.2.1 for read assembly. Blast databases were generated with the NCBI blast+-suite and screened with tblastn (RRID:SCR_011822) for opsins with a query set of 38 opsin sequences representing all of the major opsin sub-groups reported in *Hering and Mayer (2014)* and *Ramirez et al. (2016)*. Only sequences that

exhibited the PFAM 7tm_1 domain, contained Lys296 and yielded c-opsins, ctenopsins, cnidops, anthozoa II opsins *sensu Hering and Mayer (2014)* and xenopsins amongst best hits were retained. The SMEDG Unigene dataset available at the *Schmidtea mediterranea* Genome Database (RRID: SCR_007934) was screened with the same search strategy. Other opsin types, such as r-opsins, Go-opsins, RGR/peropsins, and the recently described chaopsins and bathyopsins were sampled to a lesser degree to reduce computation time of downstream analyses. As the choice of outgroup has been shown to be critical for opsin tree inference (*Feuda et al., 2012*; *Plachetzki et al., 2007*), we used a broad sampling of opsin sequences as queries to search for closely related GPCRs. This led to an outgroup consisting of melatonin, octopamine, dopamine and adrenergic receptors, and pla-copsins similar to those in the analysis of *Hering and Mayer (2014)* and more diverse than those in the analyses of *Feuda et al. (2012*, *2014)* and *Ramirez et al. (2016)*. Sequences were first aligned with MAFFT 7 with option E-INS-i. Then, ambiguously aligned N- and C-terminal regions were trimmed, sequences shorter than 160 amino acids removed and the remaining sequences again aligned with MAFFT 7 with option E-INS-I. The output was manually edited to remove gappy regions and ambiguously aligned positions, yielding an alignment of 282 positions and 258 sequences. *Feuda et al. (2012*, *2014)* and *Hering and Mayer (2014)* showed that data-set-specific substitution matrices can fit opsin alignments better than precomputed models implemented in tree inference software. Thus, we used RAxML to generate such a dataset-specific substitution matrix (DS-GTR) by parameter optimization of the general time reversible (GTR) model to the trimmed alignment and a precomputed parsimony tree. In ProtTest 3.4, the implemented model FLU was exchanged against our DS-GTR, revealing DS-GTR+$\Gamma$ as best fitting model according to Akaike information, Bayesian information and Decision theory criteria (*Figure 6—figure supplement 1*). For maximum likelihood analyses, RAxML 8.2 was run on the CIPRES Science Gateway (*Miller et al., 2010*). A first run was conducted with DS-GTR+$\Gamma$ and 500 rapid bootstraps, yielding low support for basal branching patterns. Hence, a leaf-stability analysis on the bootstrap tree set was conducted with phyutility 2.2 and a few sequences with low leaf stability were excluded from the final data-set used for all following analysis.

A new trimmed alignment was generated from the final data-set as described above and a new RAxML analysis with DS-GTR+$\Gamma$and 1000 rapid bootstraps was conducted. For Bayesian analysis, first a 10-fold cross-validation was run with phylobayes 4.1 to choose the most appropriate combination of GTR, LG and DS_GTR with non-parametric (CAT) or parametric ($\Gamma$) modeling of among-site variation or empirical mixture modeling (C20, C30, C40, C50, C60, WLSR5). As a result, DS-GTR CAT and GTR CAT were found to be the best non-parametric variants, performing better than the best parametric variant DS-GTR $\Gamma$, and all three of these were better than any empirical mixture modeling (*Figure 6*). For each of the models DS-GTR CAT, GTR CAT and DS-GTR $\Gamma$, three chains with 30,000 cycles were run, and for each, chain convergence and mixing behavior was evaluated with the tools tracecomp from phylobayes and Tracer from Beast (RRID:SCR_010228) and a burn-in of 6,000 cycles. Both non-parameteric variants yielded low estimated sample sizes for several model components suggesting poor mixing behavior, which is a known phenomenon of non-parametric models applied to small sequence alignments. Thus, only the chains with DS-GTR $\Gamma$, which all yielded high estimated sample sizes, were continued for another 30,000 cycles. Two chains, which exhibited best phylogenetic convergence assessed with bpcomp (12,000 cycles burn-in, mean deviation 0.108), were used to compute the final consensus tree.

## Opsin gene cloning from genomic DNA

For two opsins (*Platynereis dumerilii* ciliary opsin (AAV63834.1) and *Leptochiton asellus* xenopsin) the whole genes were cloned from genomic DNA for subsequent analysis of exon-intron boundaries. Genomic DNA was extracted with the Nucleospin Tissue Kit (Machery-Nagel, Düren, Germany) and tested for fragment lengths larger than 20 kb. As a starting point, gene-specific primers were designed on the basis of the transcript sequences. For genome walking, four libraries were prepared with the Universal Genome Walker Kit (Takara Bio, Kusatsu, Japan) by enzymatic digestion and used for sequence elongation starting from exonic fragments. In parallel, long amplicons bridging smaller introns were also directly amplified from genomic DNA using Lataq (Takara Bio), iProof (Biorad, Hercules, California) and HotStarTaq Plus (Qiagen, Hilden, Germany, RRID:SCR_008539) polymerases. Obtained amplicons of up to 8 kb were cloned using a pGem-T easy Vector (Promega) TOPO XL PCR cloning kit (Thermo Fisher Scientific), TopTen chemically competent cells (Thermo Fisher

| models compared | mean score ± stdev | #times model is best |
|---|---|---|
| LG CAT versus GTR CAT | -3.294 ± 11.0799 | 0 |
| LG Γ versus GTR CAT | -212.007 ± 57.1035 | 0 |
| **DS-GTR CAT versus GTR CAT** | **17.14 ± 17.4412** | **8** |
| DS-GTR Γ versus GTR CAT | -143.86 ± 44.8443 | 0 |
| LG CAT versus DS-GTR Γ | 140.566 ± 46.1317 | 0 |
| LG Γ versus DS-GTR Γ | -68.147 ± 21.4174 | 0 |
| DS-GTR CAT versus DS-GTR Γ | 161 ± 48.2159 | 8 |
| GTR CAT versus DS-GTR Γ | 143.86 ± 44.8443 | 2 |
| LG CAT versus DS-GTR CAT | -20.434 ± 17.1572 | 0 |
| LG Γ versus DS-GTR CAT | -229.147 ± 64.6082 | 0 |
| DS-GTR Γ versus DS-GTR CAT | -161 ± 48.2159 | 0 |
| GTR CAT versus DS-GTR CAT | -17.14 ± 17.4412 | 2 |
| LG Γ versus LG CAT | -208.713 ± 58.0871 | 0 |
| **DS-GTR CAT versus LG CAT** | **20.434 ± 17.1572** | **8** |
| DS-GTR Γ versus LG CAT | -140.566 ± 46.1317 | 0 |
| GTR CAT versus LG CAT | 3.294 ± 11.0799 | 2 |
| LG CAT versus LG Γ | 208.713 ± 58.0871 | 0 |
| DS-GTR CAT versus LG Γ | 229.147 ± 64.6082 | 8 |
| **DS-GTR Γ versus LG Γ** | **68.147 ± 21.4174** | **0** |
| GTR CAT versus LG Γ | 212.007 ± 57.1035 | 2 |

**Figure 6.** Ten-fold cross-validation (Phylobayes) to find best possible model for Bayesian opsin tree inference. Only some of the data are shown. Combinations based on the data set specific substitution matrix (DS-GTR) are superior to all software implemented options. DS-GTR CAT and GTR CAT are the best non-parametric variants and DS-GTR Γ is the best parametric variant. See *Figure 6—figure supplement 1* for model test for maximum-likelihood analysis and *Figure 6—source data 1* for the whole data set.

DOI: https://doi.org/10.7554/eLife.23435.021

The following source data and figure supplements are available for figure 6:

**Source data 1.** Full results of ten-fold cross-validation (Phylobayes) to find best possible model for Bayesian opsin tree inference.

DOI: https://doi.org/10.7554/eLife.23435.023

**Figure 6-figure supplement 1.** Model test (Prottest) to find the best possible model for maximum-likelihood analysis.

DOI: https://doi.org/10.7554/eLife.23435.022

**Figure supplement—source data 1.** Full results of Prottest analysis to find the best possible model for maximum-likelihood analysis.

DOI: https://doi.org/10.7554/eLife.23435.024

Scientific) and Sanger sequenced. Obtained sequences were used to design further primers for ongoing sequence elongation. Read assembly was performed with CLC Main Workstation (RRID: SCR_000354) 7.1.

## Opsin gene structure analysis

WebScipio (*Hatje et al., 2011*) was used to determine the gene structure including the respective intron phases of all those opsin sequences included in our phylogenetic analysis for which genomic data were available (1) at the WebScipio portal, (2) from other public databases (Kumamushi genome project in case of *Ramazzotius varieornatus*, or 3) as we cloned the whole genes from genomic DNA (*Platynereis dumerilii* c-opsin and *Leptochiton asellus* opsin). The obtained gene structures were than mapped onto the un-curated sequence alignment using Genepainter (*Hammesfahr et al., 2013*) in order to identify homologous intron positions and to analyze opsin gene structure

conservation. To evaluate the stability of the observed patterns against alignment artifacts, gene structures were also mapped onto additional alignments computed with MAFFT (option G-INS-I), ProbCons (RRID:SCR_011813), Muscle and GLProbs (RRID:SCR_002739).

## In-situ hybridization

Experiments were performed as described previously (*Vöcking et al., 2015*). In brief, animals were fixed in 4% PFA in phosphate buffer and with Tween20 (PTW; pH 7.4) and subsequently washed and stored in methanol. After rehydration in PTW, samples were briefly digested with Proteinase K, washed and prehybridized in hybridization buffer with 5% Dextran. Samples were hybridized with RNA probes for 72 hr and stained with a combination of FastBlue (Sigma-Aldrich, St. Louis, Missouri, RRID:SCR_008988) and Fast Red (Roche). The significance of expression signals was evaluated by using sense probes as control experiments. All in-situ hybridization experiments were performed on at least 30 specimens per gene for each sense and anti-sense probe and the experiments were repeated at least twice.

## Light microscopy

Light microscopic images were taken using Eclipse E800 (Nikon, Tokyo, Japan) and AZ100M (Nikon) microscopes and adjusted with Adobe (Mountain View, California) Photoshop (RRID:SCR_014199) CS5. Confocal images were taken with a SP5 confocal microscope (Leica, Wetzlar, Germany, RRID: SCR_008960) and the image stacks processed with ImageJ (RRID:SCR_003070) and Adobe Photoshop CS5.

## Electron microscopy

Experiments were performed as previously described (*Vöcking et al., 2015*). Briefly, animals were fixed in 2–5% glutardialdehyde in sodium cacodylate buffer, subsequently postfixed in 1% osmium tetroxide, en-bloc stained with reduced osmium, dehydrated in a graded ethanol series and embedded in Epon/Araldite. Serial sections of 70 nm were cut with an ultra 35° diamond knife (Diatome, Biel, Switzerland) on an UC7 ultramicrotome (Leica, Wetzlar, Germany) and collected on Beryllium-Copper slot grids (Synaptek, Reston, Virginia) coated with Pioloform (Ted Pella, Redding, California) and counterstained with 2% uranyl acetate and lead citrate. Complete series were imaged with STEM-in-SEM as described by *Kuwajima et al. (2013)* at a resolution of 4 nm/ pixel in a Supra 55VP (Zeiss, Oberkochen, Germany) equipped with Atlas (Zeiss) for automated large field of view imaging. Acquired images were processed with Photoshop CS5 (Adobe), first registered rigidly followed by affine and elastic alignment (*Saalfeld et al., 2012*) with TrakEM2 (RRID:SCR_008954, *Cardona et al., 2012*) implemented in Fiji (RRID:SCR_002285).

## Acknowledgements

We thank Tomas H Sørlie for his help during field sampling of *L. asellus* and Egil S Erichsen for technical help with electron microscopic imaging.

## Additional information

### Funding

| Funder | Author |
| --- | --- |
| Universitetet i Bergen | Harald Hausen |
| Norges Forskningsråd | Harald Hausen |

The funders had no role in study design, data collection and interpretation, or the decision to submit the work for publication.

### Author contributions

Oliver Vöcking, Data curation, Investigation, Visualization, Methodology, Writing—original draft, Writing—review and editing; Ioannis Kourtesis, Investigation, Visualization, Methodology; Sharat

Chandra Tumu, Resources, Investigation, Methodology; Harald Hausen, Conceptualization, Data curation, Software, Supervision, Validation, Investigation, Visualization, Methodology, Writing—original draft, Project administration, Writing—review and editing

### Author ORCIDs
Harald Hausen, http://orcid.org/0000-0003-2788-2835

### Decision letter and Author response
Decision letter https://doi.org/10.7554/eLife.23435.120
Author response https://doi.org/10.7554/eLife.23435.121

# Additional files

### Supplementary files
• Transparent reporting form
DOI: https://doi.org/10.7554/eLife.23435.025

### Major datasets
The following datasets were generated:

| Author(s) | Year | Dataset title | Dataset URL | Database, license, and accessibility information |
|---|---|---|---|---|
| Vöcking O, Kourtesis I, Tumu S, Hausen H | 2017 | Leptochiton asellus xenopsin gDNA | https://www.ncbi.nlm.nih.gov/nuccore/MF133514 | Publicly available at NCBI Nucleotide (accession no: MF133514) |
| Vöcking O, Kourtesis I, Tumu S, Hausen H | 2017 | Platynereis dumerilii ciliary-opsin gDNA | https://www.ncbi.nlm.nih.gov/nuccore/MF133515 | Publicly available at NCBI Nucleotide (accession no: MF133515) |
| Vöcking O, Kourtesis I, Tumu S, Hausen H | 2017 | Owenia fusiformis xenopsin1 cds | https://www.ncbi.nlm.nih.gov/nuccore/MF133511 | Publicly available at NCBI Nucleotide (accession no: MF133511) |
| Vöcking O, Kourtesis I, Tumu S, Hausen H | 2017 | Owenia fusiformis xenopsin2 cds | https://www.ncbi.nlm.nih.gov/nuccore/MF133512 | Publicly available at NCBI Nucleotide (accession no: MF133512) |
| Vöcking O, Kourtesis I, Tumu S, Hausen H | 2017 | Owenia fusiformis xenopsin3 cds | https://www.ncbi.nlm.nih.gov/nuccore/MF133513 | Publicly available at NCBI Nucleotide (accession no: MF133513) |
| Vöcking O, Kourtesis I, Tumu S, Hausen H | 2017 | Leptochiton asellus ADP ribosylation factor 4A cds | https://www.ncbi.nlm.nih.gov/nuccore/MF133503 | Publicly available at NCBI Nucleotide (accession no: MF133503) |
| Vöcking O, Kourtesis I, Tumu S, Hausen H | 2017 | Leptochiton asellus ADP ribosylation factor 4B cds | https://www.ncbi.nlm.nih.gov/nuccore/MF133504 | Publicly available at NCBI Nucleotide (accession no: MF133504) |
| Vöcking O, Kourtesis I, Tumu S, Hausen H | 2017 | Leptochiton asellus Rab11 family-interacting protein 3 | https://www.ncbi.nlm.nih.gov/nuccore/MF133505 | Publicly available at NCBI Nucleotide (accession no: MF133505) |
| Vöcking O, Kourtesis I, Tumu S, Hausen H | 2017 | Leptochiton asellus regulatory factor X cds | https://www.ncbi.nlm.nih.gov/nuccore/MF133506 | Publicly available at NCBI Nucleotide (accession no: MF133506) |
| Vöcking O, Kourtesis I, Tumu S, Hausen H | 2017 | Leptochiton asellus Forkhead box J1 cds | https://www.ncbi.nlm.nih.gov/nuccore/MF133507 | Publicly available at NCBI Nucleotide (accession no: MF133507) |

| Vöcking O, Kourtesis I, Tumu S, Hausen H | 2017 | Leptochiton asellus myosin VIIa cds | https://www.ncbi.nlm.nih.gov/nuccore/MF133508 | Publicly available at NCBI Nucleotide (accession no: MF133508) |
|---|---|---|---|---|
| Vöcking O, Kourtesis I, Tumu S, Hausen H | 2017 | Leptochiton asellus retinitis pigmentosa GTPase regulator cds | https://www.ncbi.nlm.nih.gov/nuccore/MF133509 | Publicly available at NCBI Nucleotide (accession no: MF133509) |
| Vöcking O, Kourtesis I, Tumu S, Hausen H | 2017 | Leptochiton asellus Ras-related protein Rab-8 | https://www.ncbi.nlm.nih.gov/nuccore/MF133510 | Publicly available at NCBI Nucleotide (accession no: MF133510) |
| Vöcking O, Kourtesis I, Tumu S, Hausen H | 2017 | Echinoplana celerrima xenopsin | https://www.ncbi.nlm.nih.gov/nuccore/BK010203 | Publicly available at NCBI Nucleotide (accession no: BK010203) |
| Vöcking O, Kourtesis I, Tumu S, Hausen H | 2017 | Prostheceraeus vittatus xenopsin A | https://www.ncbi.nlm.nih.gov/nuccore/BK010204 | Publicly available at NCBI Nucleotide (accession no: BK010204) |
| Vöcking O, Kourtesis I, Tumu S, Hausen H | 2017 | Stylochus ellipticus xenopsin A | https://www.ncbi.nlm.nih.gov/nuccore/BK010205 | Publicly available at NCBI Nucleotide (accession no: BK010205) |
| Vöcking O, Kourtesis I, Tumu S, Hausen H | 2017 | Prosthiostomum siphunculus xenopsin | https://www.ncbi.nlm.nih.gov/nuccore/BK010206 | Publicly available at NCBI Nucleotide (accession no: BK010206) |
| Vöcking O, Kourtesis I, Tumu S, Hausen H | 2017 | Prostheceraeus vittatus xenopsin B1 | https://www.ncbi.nlm.nih.gov/nuccore/BK010207 | Publicly available at NCBI Nucleotide (accession no: BK010207) |
| Vöcking O, Kourtesis I, Tumu S, Hausen H | 2017 | Prostheceraeus vittatus xenopsin B2 | https://www.ncbi.nlm.nih.gov/nuccore/BK010208 | Publicly available at NCBI Nucleotide (accession no: BK010208) |
| Vöcking O, Kourtesis I, Tumu S, Hausen H | 2017 | Stylochus ellipticus xenopsin BA | https://www.ncbi.nlm.nih.gov/nuccore/BK010209 | Publicly available at NCBI Nucleotide (accession no: BK010209) |
| Vöcking O, Kourtesis I, Tumu S, Hausen H | 2017 | Lineus longissimus GO-opsin | https://www.ncbi.nlm.nih.gov/nuccore/BK010210 | Publicly available at NCBI Nucleotide (accession no: BK010210) |

The following previously published datasets were used:

| Author(s) | Year | Dataset title | Dataset URL | Database, license, and accessibility information |
|---|---|---|---|---|
| Andrade SC, Montenegro H, Strand M, Schwartz ML, Kajihara H, Norenburg JL, Turbeville JM, Sundberg P, Giribet G | 2014 | Baseodiscus unicolor transcriptome | https://trace.ncbi.nlm.nih.gov/Traces/sra/?run=SRR1505175 | Publicly available at NCBI Sequence Read Archive (accession no: SRR1505175) |
| Andrade S | 2012 | Cephalothrix hongkongiensis transcriptome | https://trace.ncbi.nlm.nih.gov/Traces/sra/?run=SRR618505 | Publicly available at NCBI Sequence Read Archive (accession no: SRR618505) |

| | | | | |
|---|---|---|---|---|
| Andrade SC, Montenegro H, Strand M, Schwartz ML, Kajihara H, Norenburg JL, Turbeville JM, Sundberg P, Giribet G | 2014 | Hubrechtella ijimai transcriptome | https://trace.ncbi.nlm.nih.gov/Traces/sra/?run=SRR1505100 | Publicly available at NCBI Sequence Read Archive (accession no: SRR1505100) |
| Cannon JT, Vellutini BC, Smith J 3rd, Ronquist F, Jondelius U, Hejnol A | 2016 | Transcriptome of the nemertean Lineus longissimus | https://trace.ncbi.nlm.nih.gov/Traces/sra/?run=SRR2682192 | Publicly available at NCBI Sequence Read Archive (accession no: SRR2682192) |
| Halanych KM, Kocot KM | 2014 | Malacobdella grossa transcriptome | https://trace.ncbi.nlm.nih.gov/Traces/sra/?run=SRR1611560 | Publicly available at NCBI Sequence Read Archive (accession no: SRR1611560) |
| Andrade SC, Montenegro H, Strand M, Schwartz ML, Kajihara H, Norenburg JL, Turbeville JM, Sundberg P, Giribet G | 2014 | Nipponnemertes transcriptome | https://trace.ncbi.nlm.nih.gov/Traces/sra/?run=SRR1508368 | Publicly available at NCBI Sequence Read Archive (accession no: SRR1508368) |
| Kocot K | 2014 | Paranemertes peregrina transcriptome | https://trace.ncbi.nlm.nih.gov/Traces/sra/?run=SRR1611562 | Publicly available at NCBI Sequence Read Archive (accession no: SRR1611562) |
| Whelan N | 2014 | Transcriptome sequencing of Tubulanus polymorphus | https://trace.ncbi.nlm.nih.gov/Traces/sra/?run=SRR1611583 | Publicly available at NCBI Sequence Read Archive (accession no: SRR1611583) |
| Egger B | 2015 | Echinoplana celerrima | https://trace.ncbi.nlm.nih.gov/Traces/sra/?run=SRR1796488 | Publicly available at NCBI Sequence Read Archive (accession no: SRR1796488) |
| Laumer C | 2015 | Geocentrophora applanata transcriptome | https://trace.ncbi.nlm.nih.gov/Traces/sra/?run=SRR1955490 | Publicly available at NCBI Sequence Read Archive (accession no: SRR1955490) |
| Egger B | 2015 | Itaspiella helgolandica | https://trace.ncbi.nlm.nih.gov/Traces/sra/?run=SRR1797713 | Publicly available at NCBI Sequence Read Archive (accession no: SRR1797713) |
| Laumer C | 2015 | Kronborgia cf. amphipodicola transcriptome | https://trace.ncbi.nlm.nih.gov/Traces/sra/?run=SRR1976457 | Publicly available at NCBI Sequence Read Archive (accession no: SRR1976457) |
| Egger B | 2015 | Microdalyellia schmidti | https://trace.ncbi.nlm.nih.gov/Traces/sra/?run=SRR1797778 | Publicly available at NCBI Sequence Read Archive (accession no: SRR1797778) |
| Laumer C | 2015 | Prorhynchus sp. I transcriptome | https://trace.ncbi.nlm.nih.gov/Traces/sra/?run=SRR1980634 | Publicly available at NCBI Sequence Read Archive (accession no: SRR1980634) |
| Laumer C | 2015 | Prostheceraeus vittatus transcriptome | https://trace.ncbi.nlm.nih.gov/Traces/sra/?run=SRR2000268 | Publicly available at NCBI Sequence Read Archive (accession no: SRR2000268) |
| Egger B | 2015 | Prosthiostomum siphunculus | https://trace.ncbi.nlm.nih.gov/Traces/sra/?run=SRR1797833 | Publicly available at NCBI Sequence Read Archive (accession no: SRR1797833) |
| Egger B | 2015 | Stenostomum sthenum | https://trace.ncbi.nlm.nih.gov/Traces/sra/?run=SRR1801788 | Publicly available at NCBI Sequence Read Archive (accession no: SRR1801788) |

| Laumer C | 2015 | Stylochus ellipticus larvae transcriptome | https://trace.ncbi.nlm.nih.gov/Traces/sra/?run=SRR1980690 | Publicly available at NCBI Sequence Read Archive (accession no: SRR1980690) |
|---|---|---|---|---|
| Cannon JT, Vellutini BC, Smith J 3rd, Ronquist F, Jondelius U, Hejnol A | 2016 | Transcriptome of the nemertodermatid Ascoparia sp. | https://trace.ncbi.nlm.nih.gov/Traces/sra/?run=SRR2682154 | Publicly available at NCBI Sequence Read Archive (accession no: SRR2682154) |
| Cannon JT, Vellutini BC, Smith J 3rd, Ronquist F, Jondelius U, Hejnol A | 2016 | Transcriptome of the acoel Diopisthoporus longitubus | https://trace.ncbi.nlm.nih.gov/Traces/sra/?run=SRR3105704 | Publicly available at NCBI Sequence Read Archive (accession no: SRR3105704) |
| Cannon JT, Vellutini BC, Smith J 3rd, Ronquist F, Jondelius U, Hejnol A | 2016 | Transcriptome of the acoel Eumecynostomum macrobursalium | https://trace.ncbi.nlm.nih.gov/Traces/sra/?run=SRR3105705 | Publicly available at NCBI Sequence Read Archive (accession no: SRR3105705) |
| Cannon JT, Vellutini BC, Smith J 3rd, Ronquist F, Jondelius U, Hejnol A | 2016 | Transcriptome of the acoel Isodiametra pulchra | https://trace.ncbi.nlm.nih.gov/Traces/sra/?run=SRR2681926 | Publicly available at NCBI Sequence Read Archive (accession no: SRR2681926) |
| Cannon JT, Vellutini BC, Smith J 3rd, Ronquist F, Jondelius U, Hejnol A | 2016 | Transcriptome of the nemertodermatid Meara stichopi | https://trace.ncbi.nlm.nih.gov/Traces/sra/?run=SRR2681155 | Publicly available at NCBI Sequence Read Archive (accession no: SRR2681155) |
| Cannon JT, Vellutini BC, Smith J 3rd, Ronquist F, Jondelius U, Hejnol A | 2016 | Transcriptome of the nemertodermatid Nemertoderma westbladi | https://trace.ncbi.nlm.nih.gov/Traces/sra/?run=SRR2682004 | Publicly available at NCBI Sequence Read Archive (accession no: SRR2682004) |
| Cannon JT, Vellutini BC, Smith J 3rd, Ronquist F, Jondelius U, Hejnol A | 2016 | Transcriptome of the nemertodermatid Sterreria sp. | https://trace.ncbi.nlm.nih.gov/Traces/sra/?run=SRR2682099 | Publicly available at NCBI Sequence Read Archive (accession no: SRR2682099) |
| Cannon JT, Vellutini BC, Smith J 3rd, Ronquist F, Jondelius U, Hejnol A | 2016 | Transcriptome of Xenoturbella bocki | https://trace.ncbi.nlm.nih.gov/Traces/sra/?run=SRR2681987 | Publicly available at NCBI Sequence Read Archive (accession no: SRR2681987) |

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
