## [Decision Letter]

Thank you for submitting your article "A ciliary opsin-like pigment co-expressed with rhabdomeric opsin in mixed rhabdomeric/ciliary photoreceptors" for consideration by *eLife*. Your article has been reviewed by two peer reviewers, and the evaluation has been overseen by a Reviewing Editor and Eve Marder as the Senior Editor. The following individuals involved in review of your submission have agreed to reveal their identity: Todd Oakley (Reviewer #3).

The reviewers have discussed the reviews with one another and the Reviewing Editor has drafted this decision to help you prepare a revised submission.

General Comments:

The reviewers agreed that this paper provides useful and interesting new information relevant to the resolution of opsin evolution. This work reports that a xenopsin is expressed in the larval photoreceptor of a chiton which also express r-opsin and that the cells have not only microvilli, but also cilia, in addition to expressing genetic markers for "targeting of opsin into cilia." These data may help explain why some taxa have photoreceptor cells with both cilia and microvilli and others do not. Xenopsin clearly includes cnidops, and so this group of opsins is very old – dating at least to the common ancestor of cnidarians and bilaterians.

However, the reviewers agree that:

1) The writing must be improved dramatically. This will help make the paper understandable to those outside the evolution of vision field. The writing must be in more standard, grammatical English that will only improve the impact of their work. They need to distill the message and more specifically state the overall importance of the work early on in the manuscript. In addition, they need to incorporate recent work by Ramirez et al. who already named the "col" group as "xenopsins" and came to several of the same conclusions about the distribution of these opsins. This will allow the authors to really highlight the important results they found on co-expression and microscopy.

2) The unique analysis including intron position and phase, of the opsins reveals two group-specific introns that are conserved in position and phase, as opposed to three unrelated introns in the c-opsin group, and thus lends strong support to the coherence of the Lophotrochozoan col-opsins. This point should be emphasized.

Other important points to be addressed:

1) Please add accession numbers to the names of the genes. This is important in Figure 1 and in supplemental figures to allow future researchers to repeat/recheck the analyses.

2) The first part of the Materials and methods do not state which substitution model they used (just states 'γ' – looks like they state DS-GTR later). How did you test for convergence of the mcmc? Which leaf stability analysis? The authors must supply accession numbers, ideally directly on the phylogeny in the supplement. What does a "broad and balanced" gene sampling mean (this statement is arbitrary)?

3) Additional data files and statistical comments:

It would be very useful to be rigorous about Supplementary files. Speaking from extensive experience here, the authors should include accession numbers for all their sequences. Ideally these would be in the figures of all the trees (as part of the gene names); and perhaps also on a table with species and accession numbers.

4) Regarding the molecular phylogeny, a few important animal groups that are not mentioned, even though they can be mined and would likely be informative. These are the nemertines, the priapulids, and the acoels. They are important because each of them might possess both opsins, and thus would be a good test case for the reciprocal loss paradigm (either the one or the other). This should be mentioned and reasons for exclusion discussed.

5) What are the specific sequence similarity to the c-opsins? Do they go beyond the NKQ motif? If yes, this should be documented.

6) The link between cnidarian and bilaterian opsin groups still appears to be somewhat arbitrary. It appears that there are at least 3 old opsin subgroups, but the link to the bilaterian groups appears to change from paper to paper and is weak at best. Adding to this, the intron analysis does not support a link between cnidopsin and colopsin. This should be mentioned and the possibility that colopsins and cnidopsins are not related should be emphasized. This does not take away from the strength of the paper. Please mention this possibility also in the interpretative scheme (see below).

7) Relating to this, it should be mentioned that the cnidarian anthozoan opsin II group is related to c-opsins by one intron with similar position (Figure 2—figure supplement 2) and in phase (Figure 2—figure supplement 3), which is remarkable, as it strengthens orthology of cnidarian anthozoan II with bilaterian ciliary opsins as postulated by Ramirez et al. This should be highlighted in the manuscript, the more so since it shows that intron conservation between cnidarians and ctenophores is possible.

8) Also, regarding the intron analysis, the 2 introns of the colopsin group may still find counterparts in the c-opsin and opsin-4 groups, if the protein alignment is slightly "corrected" by a few positions. How strong is the AA alignment of the small bits of AA sequence between these intron positions? Alternatively, why are these seemingly different positions not only slight shifts of one and the same intron?

9) Even though somewhat hidden in the literature, there is a lot of evidence that motile cilia indeed belong to the ancient 'equipment ' of rhabdomeric photoreceptor cells, and the authors should discuss this. This has long been stated and is not a novel finding. The senior author has published himself an evolutionary scenario on 'division of labor' between motile ciliated cells and rhabdomeric photoreceptor cells, in line with this. Thus, it is by no means certain that this is a 'ciliary' photoreceptor. The presence of a cilium is rather neutral with regard to this question.

11) Also, it should be stated that the 'transporters known to be involved in targeting opsins into cilia' are in fact more general transport proteins for primary cilia. How can the authors rule out that they are transporting something else? This should be mentioned and discussed. How about the specific ciliary localization signals known to be recognized by these proteins, as mentioned for example in Wang and Deretic 2014? If these signals are not present in the colopsins, this casts doubt on their ciliary transport. The only clear evidence for ciliary localization would be immunolocalization. In the absence of these data, the point remains weak and should be discussed as such.

12) The two interpretation schemes are almost the same. I would propose to unite them both into one scheme (with two variants and question marks) and include this in the main paper. Then, I would urge the authors to also include the possibility that col-opsin is Lophotrochozoan-specific (see my comments above, taking into account that the orthology to cnidopsin is still tentative.) The green colopsin line should thus be dashed outside Lophotrochozoans.

[Editors' note: further revisions were requested prior to acceptance, as described below.]

Thank you for resubmitting your article "Co-expression of xenopsin and rhabdomeric opsin in photoreceptors bearing microvilli and cilia" for consideration by *eLife*. Your article has been reviewed by two peer reviewers, and the evaluation has been overseen by a Reviewing Editor and Eve Marder as the Senior Editor.

The reviewers have discussed the reviews with one another and the Reviewing Editor has drafted this decision to help you prepare a revised submission. I think the authors have carefully considered all the scientific points raised by the reviewers. I think the paper in much improved in that respect, with the importance, and uniqueness showing more. We hope you will be able to deal with the further textual changes listed below and return a final revised version as quickly as possible.

Summary:

The evolution of opsin, the seven-transmembrane domain protein that collaborates with a chromophore to transform photons into electrical signals, preceded the evolution of eyes. In extant animals, opsins located in eyes in cilia and rhabdomeres are used by cells differing fundamentally in structure and molecular physiology, such as ciliary vertebrate rods and cones and protostome microvillar eye photoreceptors. The authors report unprecedented cellular co-expression of rhabdomeric opsin and a visual pigment of the recently described xenopsins in larval eyes of a mollusk. The photoreceptors have both microvilli and cilia and express transporters for microvillar and ciliary opsin trafficking. Gene structure shows that xenopsin and ciliary opsins are of independent origin irrespective of their mutually exclusive distribution in animals. The authors propose that xenopsins are ancient components of animal eyes and initially may have been employed by a highly plastic photoreceptor cell type of mixed microvillar/ciliary organization. Frequent secondary loss of opsins likely played a key role in the early evolution of animal photoreceptors influencing the organization and function of photoreceptor cells found in brains and eyes extant animals.

Essential revisions:

The writing can be improved dramatically by focusing on streamlining, by focusing on style, and by focusing on grammar. Here are specific suggestions:

1) Justify the choice to make the non-opsin outgroups monophyletic, in contrast to previous results (especially Feuda et al. PNAS, 2012)?

Abstract:

Please avoid the word "basal" to describe a living species (as in "larval eyes of a basal mollusk").

"Transporters for opsin". To be precise you found homologs of proteins known in other species to be transporters.

"today living animals" is not grammatical. You mean "extant animals".

Introduction:

"in times," is not grammatical. I think you can remove it.

First paragraph; Photoreceptors are not the same as eyes. I suppose you might be discussing old debates, but these lines make it seem like PRC=eye.

Second paragraph; "upcoming" is the wrong word. You mean "Subsequent".

Second paragraph; Evolution-based (needs hyphen). Remove the word "meanwhile", it is not grammatical as written.

Second paragraph, last sentence; Not grammatical. This is a key sentence, where the authors are trying to state why a general reader should care about all the detail they are laying out. This important sentence needs to make sense, and be grammatical. (and probably should not be the last sentence of a paragraph).

Third paragraph; "enrolled" is wrong word. "are involved in", perhaps.

Third paragraph; "In consequence". Not grammatical.

Last paragraph; "independent evolutionary origin". Misleading. They are both opsins. What you mean is they are paralogs, not orthologs.

Last paragraph; Living animals are not "basal".

Last paragraph; What does "signaling mode" mean?

Results:

"screening own" not grammatical

Subsection “A new xenopsin from the chiton *Leptochiton asellus*”, first paragraph; "a few cnidops". The original name is "cnidops" not "cnidopsin". You also need the "a" to get across what you mean.

"Since the distinction between xenopsins and c-opsins is so far only based on maximum- likelihood tree inference, we performed both maximum-likelihood and Bayesian analyses." This is not a logical statement. By this logic – you should also perform Parsimony, Neighbor Joining, and random tree generation. Furthermore, a lot of this is methods. Why is it in the Results section?

Subsection “A new xenopsin from the chiton *Leptochiton asellus*”, first paragraph; What special care?

Subsection “A new xenopsin from the chiton *Leptochiton asellus*”, first paragraph; "Own". Do you mean newly generated data?

Subsection “A new xenopsin from the chiton *Leptochiton asellus*”, first paragraph; A living annelid is not basally branching.

Subsection “A new xenopsin from the chiton *Leptochiton asellus*”, last paragraph; "paralogs distribute within and outside of a well-supported xenopsin subgroup". Not clear what that means.

Subsection “Gene structure supports independent origins of xenopsins and c-opsins irrespective of their mutually exclusive distribution”, first paragraph; "are available" is the wrong word. "are recovered" – perhaps or "are identified" (if you want to stick with passive voice).

Subsection “Gene structure supports independent origins of xenopsins and c-opsins irrespective of their mutually exclusive distribution”, first paragraph; – The living annelid is still not basal.

Subsection “Gene structure supports independent origins of xenopsins and c-opsins irrespective of their mutually exclusive distribution”, second paragraph”, last sentence; "anyway" is not usually used this way in written English.

Subsection “Gene structure supports independent origins of xenopsins and c-opsins irrespective of their mutually exclusive distribution”, last paragraph” Again "independent" is misleading. They are both opsins, but different paralogs.

Subsection “Gene structure supports independent origins of xenopsins and c-opsins irrespective of their mutually exclusive distribution”, last paragraph; – "corroborate" -> "corroborating".

Abstract: “Gene structure shows that xenopsin and ciliary opsins are of independent origin irrespective of their mutually exclusive distribution in animals.”

Gene structures can evolve. Please tone down to:

"Highly conserved but distinct gene structures suggest that xenopsins and ciliary opsins are of independent origin, irrespective of their mutually exclusive distribution in animals."

Introduction: "By deeply mining public and new data we find xenopsins to be present in several taxa of protostome invertebrates."

Please be more precise, otherwise you suggest a broader distribution than detected: "… to be present in several taxa of protostome lophotrochozoans."

Results: "Several of the key players of ciliary opsin targeting, namely Las-Arf4, Las-rab8, Las- 194 FIP3, Las-RPGR and Las-MyosinVIIa". Again, it is more appropriate to refer to:

"several general players of ciliary targeting also active in ciliary opsin targeting…"

Discussion: I don't understand why you write "presence of xenopsin in ancestral eye PRCs seems likely". What do you mean by 'ancestral' here? Do you mean the ancestor of all bilaterians? Any ancestor (like of chitons)? Ancestor of all eyes.

Discussion:

First sentence; Retina -> retinal.

Third sentence; Grammar problem – as written, this sentence means only a few protostome sequences were known. You mean that only a few protostome c-opsins were known.

Second paragraph; “Probably due to the increased taxon sampling…” and due to formal reconciled tree analysis.

Third paragraph; basal.

Fifth paragraph; Idea – the convergent evolution of the NKQ motif made xenopsin and c-opsins use the same phototransduction cascade. This could make them redundant, and allow for only one to usually be retained.

Materials and methods:

Subsection “Opsin tree inference”, second paragraph; Should cite Cannon et al. for Xenacoelomorpha.

Subsection “Xenacoelomorpha”, second paragraph; But the papers you cite show that including distant GPCR outgroups is a problem, yet that is what you chose to do.

Figures:

Why did you decide to root the tree with all the non-opsins as monophyletic? Others have found that Trichoplax opsin is a sister group to canonical opsins. Your tree is consistent with this, but you seemed to have chosen the outgroups to be monophyletic, with no justification as to why.

---

## [Author Response]

General Comments:The reviewers agreed that this paper provides useful and interesting new information relevant to the resolution of opsin evolution. This work reports that a xenopsin is expressed in the larval photoreceptor of a chiton which also express r-opsin and that the cells have not only microvilli, but also cilia, in addition to expressing genetic markers for "targeting of opsin into cilia." These data may help explain why some taxa have photoreceptor cells with both cilia and microvilli and others do not. Xenopsin clearly includes cnidops, and so this group of opsins is very old – dating at least to the common ancestor of cnidarians and bilaterians.However, the reviewers agree that:1) The writing must be improved dramatically. This will help make the paper understandable to those outside the evolution of vision field. The writing must be in more standard, grammatical English that will only improve the impact of their work. They need to distill the message and more specifically state the overall importance of the work early on in the manuscript. In addition, they need to incorporate recent work by Ramirez et al. who already named the "col" group as "xenopsins" and came to several of the same conclusions about the distribution of these opsins. This will allow the authors to really highlight the important results they found on co-expression and microscopy.

We rewrote many passages in order to improve the English. We further restructured the Introduction and parts of the Results and Discussion in order to stress more the significant findings and statements. Following the recommendations from the reviewers we screened under-sampled taxa for opsins. This required de novo assembling of numerous publicly available RNA-seq data sets and to rerun all opsin phylogenies and gene structure analyses. Materials and methods were extended to provide more details on the tree inference as it was suggested from the reviewers. All figures on opsin phylogeny and gene structure are updated.

We discuss now the study of Ramirez in more detail and took over the term xenopsin. However, since gene structure data do not provide additional support for the proposed sistergroup relationship of cnidopsins and bilaterian xenopsins, we prefer to apply the term xenopsins only to bilaterian sequences and not to cnidopsins.

Although we refer many times to Ramirez, we have to discuss the distribution of xenopsins as well in detail. First, we found xenopsins in additional taxa and second, we are the first describing the remarkable mutually exclusive distribution of xenopsins and c-opsins.

2) The unique analysis including intron position and phase, of the opsins reveals two group-specific introns that are conserved in position and phase, as opposed to three unrelated introns in the c-opsin group, and thus lends strong support to the coherence of the Lophotrochozoan col-opsins. This point should be emphasized.

We fully agree that the performed gene structure analysis adds very valuable information for unravelling opsin evolution. We regard the observed differences in gene structure between xenopsins and c-opsins as a valuable piece of information, which is independent from tree inference and provides strong evidence that these two opsin groups are really distinct, irrespective of their mutually exclusive taxonomic distribution.

Within the text the significance of the obtained results is now stronger emphasized and we refer to the gene structure also in another context as e.g.:

- the validity of the small group of bathyopsins containing one sequence from *Strongylocentrotus*, which formerly was assumed to be allied to xenopsins

- the divergence of anthozoan opsin II and ctenopsins and the possible relationship of these opsins to c-opsins.

Other important points to be addressed:1) Please add accession numbers to the names of the genes. This is important in Figure 1 and in supplemental figures to allow future researchers to repeat/recheck the analyses.

Accession numbers are now added to all sequences presented in figures and also provided as additional data with the only exception of those sequences still awaiting approval and curation from GenBank.

2) The first part of the Materials and methods do not state which substitution model they used (just states 'γ' – looks like they state DS-GTR later). How did you test for convergence of the mcmc? Which leaf stability analysis? The authors must supply accession numbers, ideally directly on the phylogeny in the supplement. What does a "broad and balanced" gene sampling mean (this statement is arbitrary)?

We give now a much more precise description of the opsin tree inference in Materials and methods.

We describe now in detail the sequence sampling procedure, which had the aim to obtain a data-set covering c-opsins, cnidopsins, ctenopsins, xenopsins and anthozoa opsins II from very many taxa. Since we were aiming computational very extensive Bayesian analyses we reduced sequence sampling in groups like vertebrates and arthropods, where many sequences are reported and in other opsin groups, which are not in focus of the study.

We describe how a data-set specific substitution matrix (DS-GTR) was generated to apply the best possible evolutionary models in maximum-likelihood and Bayesian analyses.

-

We also describe how different models were tested (ProtTest for maximum likelihood and 10 x cross validation for Bayesian analyses) and added respective figures (Figure 6, Figure 6—figure supplement 1) and the whole datasets (Figure 6—source data 1, Figure 6—figure supplement 1—source data 1) to the manuscript.

We describe how model mixing behaviour, and how phylogenetic convergence of Bayesian runs was estimated and we provide the relevant information about the leaf stability analysis.

Accession numbers have been added to all listed sequences and are also provided as additional data with the only exception of those sequences still awaiting approval and curation from GenBank.

3) Additional data files and statistical comments:It would be very useful to be rigorous about Supplementary files. Speaking from extensive experience here, the authors should include accession numbers for all their sequences. Ideally these would be in the figures of all the trees (as part of the gene names); and perhaps also on a table with species and accession numbers.

Accession numbers have been added to all listed sequences in all figures and are also provided as additional data with the only exception of those sequences still awaiting approval and curation from GenBank. Further, we show now the metrics of the model tests performed.

4) Regarding the molecular phylogeny, a few important animal groups that are not mentioned, even though they can be mined and would likely be informative. These are the nemertines, the priapulids, and the acoels. They are important because each of them might possess both opsins, and thus would be a good test case for the reciprocal loss paradigm (either the one or the other). This should be mentioned and reasons for exclusion discussed.

Public genomic data of priapulids suggest that priapulids are devoid of opsins. No or only very few opsin sequences are available for other protostome groups such as nemerteans or acoels. Thus, we downloaded for the revised version of the manuscript raw Illumina RNA-seq data sets (SRA archive) from Genbank for 9 nemertean, 7 acoelomorph, 1 xenoturbellid and in addition 10 platyhelminth taxa and computed for all data de novo assemblies, which than were screened for opsins similar to c-opsins, xenopsins, anthozoan opsins2, cnidopsins or ctenopsins.

Indeed we found some new sequences, which add valuable information to the analysis. Further, we found one new sequence in the SMEDG database of the flatworm *Schmidtea mediterranea* and we retrieved complete sequences from the Genbank TSA archive matching the publicly available fragments of *Hypsibius dujardini* c-opsins, which than also were included into the data set.

As a consequence, we run all opsin tree and gene structure analyses anew from scratch. The whole process was computationally demanding and thus, we mention the significance of the RNA-seq screens several times in Results and Discussion. The screened data provide clear (and new) evidence for the presence of xenopsins in flatworms, but not in acoels or nemerteans.

5) What are the specific sequence similarity to the c-opsins? Do they go beyond the NKQ motif? If yes, this should be documented.

No, sequence similarity between several xenopsins and c-opsins is to our knowledge restricted to the presence of the NKQ tripeptide motif.

6) The link between cnidarian and bilaterian opsin groups still appears to be somewhat arbitrary. It appears that there are at least 3 old opsin subgroups, but the link to the bilaterian groups appears to change from paper to paper and is weak at best. Adding to this, the intron analysis does not support a link between cnidopsin and colopsin. This should be mentioned and the possibility that colopsins and cnidopsins are not related should be emphasized. This does not take away from the strength of the paper. Please mention this possibility also in the interpretative scheme (see below).

We agree that the basal branching patterns in opsin phylogenies differ considerably between studies and depend obviously strongly on taxon sampling, choice of outgroup and probably tree inference algorithms and parameters. Our phylogenetic analysis differed in many of these aspects from the study of Ramirez et al. (2016), but likewise recovered the sistergroup relationship of xenopsins and cnidopsins. However, gene structure analysis could not corroborate this grouping, since cnidposins lack introns. In difference to Ramirez et al. (2016), we thus prefer to use the adopted term xenopsins not for cnidopsins, but exclusively for sequences from bilaterians. This is now also mentioned in the manuscript.

7) Relating to this, it should be mentioned that the cnidarian anthozoan opsin II group is related to c-opsins by one intron with similar position (Figure 2—figure supplement 2) and in phase (Figure 2—figure supplement 3), which is remarkable, as it strengthens orthology of cnidarian anthozoan II with bilaterian ciliary opsins as postulated by Ramirez et al. This should be highlighted in the manuscript, the more so since it shows that intron conservation between cnidarians and ctenophores is possible.

To address this aspect in more detail we considerably enhanced the number of anthozoan opsins II included in the analyses. However, we do not see clear evidence for a close relationship of anthozoan opsin II and c-opsins. Interestingly, the increased sequence sampling shows that anthozoan opsins II consist of two different subgroups with evidence from opsin tree and gene structure. Conserved anthozoan II opsin introns, however, do not match in position and phase to any conserved c-opsin intron.

8) Also, regarding the intron analysis, the 2 introns of the colopsin group may still find counterparts in the c-opsin and opsin-4 groups, if the protein alignment is slightly "corrected" by a few positions. How strong is the AA alignment of the small bits of AA sequence between these intron positions? Alternatively, why are these seemingly different positions not only slight shifts of one and the same intron?

We agree that the first conserved xenopsin intron matches in phase and has a position close to the second conserved c-opsin intron. To evaluate, whether the slight difference in position may be due to alignment artefacts and thus mask a common evolutionary origin, we mapped the gene structures on another 4 alignments generated with other alignment tools or parameters (MAFFT- G-INS-I, GLProbs, ProbCons, MUSCLE). In no case the respective intron position were overlapping. This observation is now described in the manuscript.

For illustrations we prepared additional figures similar to Figure 2—figure supplement 2, where the gene structures are mapped on the new alignments. We were not sure, whether inclusion of so many similar figures into the manuscript would be appropriate or an overload. Thus, we did so far not include these figures, but we will be glad to provide them, if this is desired.

Intron drift describes small positional (few basepairs) changes of introns. It seems to be a rather rare phenomenon in eukaryotes (see e.g. Roy, S (2009). BMC Evol. Biol. / Bocco SS et al. (2016). Genome Biol. Evol. or Li et al. (2017), Yangtze Medicine). Nevertheless, we cannot completely rule out that the mentioned introns in xenopsins and c-opsins were not affected by intron drift. The assumption of a common origin of the first conserved xenopsin intron and the second conserved c-opsin intron would require intron drift of one of the introns in the stem lineage of either c-opsins or xenopsins. This is not impossible, but we do not see any evidence for this. Furthermore, even the assumption of a common origin of the mentioned introns, would not change the main conclusions of the study. Even then two c-opsin and one xenopsin introns are remaining, which do not match introns in the other opsin group. This does not point towards an integration of xenopsins within c-opsins or vice versa. Only a sistergroup relationship of xenopsin and c-opsins could be conceivable, although this would be in conflict with the opsin tree. Anyway, the main statement that c-opsins and xenopsins were present as distinct groups already in the last common ancestor of protostomes and deuterostomes would not change.

We do not see evidence for the presence of a conserved shared intron in xenopsins and tetraopsins. The one conserved tetraopsin intron, which is close in position to the second conserved xenopsin intron differs in intron phase. Similar to the case described above, the difference in position is retained when gene structures are mapped on other alignments.

Rather a closer relation of ctenopsins to c-opsins may be inferred from the gene structure data. According to our tree, ctenopsins diverged into two subgroups, which share some introns, but show also subgroup specific introns. One subgroup specific intron matches in phase and position with the third conserved c-opsin intron. Since only sequences from 2 ctenophore species were included in the analysis more reliable statements on this topic may require the advent of genomic resources of further ctenophore species.

9) Even though somewhat hidden in the literature, there is a lot of evidence that motile cilia indeed belong to the ancient 'equipment ' of rhabdomeric photoreceptor cells, and the authors should discuss this. This has long been stated and is not a novel finding. The senior author has published himself an evolutionary scenario on 'division of labor' between motile ciliated cells and rhabdomeric photoreceptor cells, in line with this. Thus, it is by no means certain that this is a 'ciliary' photoreceptor. The presence of a cilium is rather neutral with regard to this question.

We discuss this topic now in more detail in the manuscript. Indeed, the presence of rudimentary cilia in microvillar PRCs is well documented. It has been interpreted in different ways. 1) those cilia were assumed to be remnants of motile cilia from epidermal cells (of the neuroectoderm), which gave rise to the PRCs. 2) An inductory function of the cilia for the development of cilia was assumed, which was never confirmed by empirical data. 3) A former sensory function was only assumed in context of a hypothesis favouring multiple origins of eye PRCs and eyes in general from inconspicuous cells bearing microvilli and cilia enrolled in a diffuse dermal light sense. This view was contradicted by upcoming molecular data clearly favouring common origin of many eye PRCs and existence of only few different eye PRC types. Xenopsin has been shown to be strongly expressed in eye PRCs of purely ciliary organization (brachiopod larvae) and we find it in cells bearing microvilli and cilia and expressing transporters known to be involved in cililary opsin trafficking (see below). From these observations we draw the conclusion that the cilia of the eye PRCs of L. asellus may have a sensory function. Presence of a central pair of microtubule in the ciliary axoneme is not informative. It is required for motility and not for a sensory function, but it does not contradict a sensory function. Indeed many sensory cilia described contain a central pair of microtubules. A motile function is unlikely, since the cilia do not penetrate the cuticle. Noteworthy, the last two decades brought into light that most cells of vertebrates bear small inconspicuous cilia (primary cilia), which are sensory and fulfil highly important functions in intercellular signalling and development. Moreover, the distinction in motile and sensory cilia is meanwhile no longer regarded as clear cut. Several cases were documented, where motile cilia take also in a sensory function and many data point towards a dual function of the cilia from early eukaryotes. In consequence, a dual function sensory motile function of cilia in ancient photoreceptors is likewise conceivable.

10) Also, it should be stated that the 'transporters known to be involved in targeting opsins into cilia' are in fact more general transport proteins for primary cilia. How can the authors rule out that they are transporting something else? This should be mentioned and discussed. How about the specific ciliary localization signals known to be recognized by these proteins, as mentioned for example in Wang and Deretic 2014? If these signals are not present in the colopsins, this casts doubt on their ciliary transport. The only clear evidence for ciliary localization would be immunolocalization. In the absence of these data, the point remains weak and should be discussed as such.

Immunolocalization of the opsin certainly would add valuable data to the manuscript. Actually, we prepared several custom antibodies for this purpose, but these did not give any specific signals, what unfortunately happens quite often with custom made antibodies. Albeit we regard it as very likely that xenopsin is able to enter cilia due to its documented strong expression in the larval brachiopod ciliary eye PRCs, whose cilia give rise to a very large sensory membrane surface.

Arf4, FIP3, Rab11 and ASAP1 are well known as important actors for targeting vertebrate rhodopsin into the ciliary outer segment of rods and cones. According to the literature two motifs are known which are of high relevance for cargo binding: The C-terminal VxPx motif and the FR next to the tripeptide motif within the 8^th^ TM helix. L. asellus xenopsin is lacking the VxPx motif. This observation may be interpreted as arguing against the proposed transport of xenopsin into the cilia of the L. asellus PRCs. But, from our data it is clear that ciliary opsin targeting does not depend on the presence of the VxPx motif. By aligning the C-terminal ends of all investigated sequences it became obvious that the VxPx motif is conserved throughout vertebrate visual opsins, pinopsins, parapinopsins, VA-opsins and encephalopsins and Ciona-op1, but nowhere else.

Notably, the VxPx is absent in the c-opsins from cephalochordates (where c-opsin expression has been reported from the ciliary frontal organ), echinoderm c-opsin and all protostome c-opsins (where likewise c-opsin expression has been reported from ciliary PRCs). It is generally lacking in xenopsins (where expression is known in purely ciliary PRCs) and in cnidopsins (which also are known to enter cilia). Either all those opsins depend on completely different targeting systems (and c-opsins switched than to Arf4 mediated transport in the vertebrate lineage) or the Arf4 complex is recognizing other sequence motifs in non-vertebrate opsins.

In conclusion we suggest that standard ciliary opsin trafficking takes place in the eye PRCs of L. asellus due to the obvious general capability of xenopsins to enter cilia, the expression of the Arf4 complex genes in the PCRs of L. asellus and the fact that many opsins lacking the VxPx motif are targeted into cilia.

As requested, we address this aspect now in more detail in the manuscript.

The second motif (FR) involved in Arf4 mediated ciliary targeting of vertebrate c-opsin is highly conserved in many opsin types: c-opsins, xenopsins, cnidopsins, but also many opsins r-opsins, Go-opsins and neuropsins and as such is not informative about trafficking references. However, the FR motif forms in c-opsins part of the larger NKQFR motif, which is conserved throughout c-opsins (also non-chordate and prostome c-opsins) and in many xenopsins (including L. asellus xenopsin). This extended motif is involved in G-protein coupling during signal transduction, but maybe it is also relevant for the opsin targeting. This Certainly, this remains speculative and needs to be investigated. Thus, we do not mention this in the manuscript.

11) The two interpretation schemes are almost the same. I would propose to unite them both into one scheme (with two variants and question marks) and include this in the main paper. Then, I would urge the authors to also include the possibility that col-opsin is Lophotrochozoan-specific (see my comments above, taking into account that the orthology to cnidopsin is still tentative.) The green colopsin line should thus be dashed outside Lophotrochozoans.

We combined the two figures into one.

Although the relationship of xenopsins and cnidopsins may need further investigation, we see strong evidence that xenopsins were already present in the bilaterian ancestor. Otherwise xenopsins would have to be the sistergroup to another protostome specific subgroup of opsins. And such an assumption would require major rearrangements of the tree topology and thus is regarded as a very unlikely scenario.

[Editors' note: further revisions were requested prior to acceptance, as described below.]

Essential revisions:The writing can be improved dramatically by focusing on streamlining, by focusing on style, and by focusing on grammar. Here are specific suggestions:1) Justify the choice to make the non-opsin outgroups monophyletic, in contrast to previous results (especially Feuda et al. PNAS, 2012)?

We admit that we did not pay much attention on outgroup topology when rooting the trees, since the interrelationships of the outgroup GPCRs are not relevant for any conclusion made and are not discussed in the manuscript. In accordance with Feuda et al. (2012), who analysed outgroup interrelationships in more detail, we now root all trees in a manner that melatonin receptors and Trichoplax opsin-like sequences are the closest relatives to animal opsins. Respective changes were made in Figure 2 and Figure 2—figure supplement 1. In addition, the order of outgroup sequences was changed in Figure 1—figure supplement 1, Figure 2—figure supplement 2, Figure 2—figure supplement 3 and Figure 4—figure supplement 2. The changed root position does not in any sense affect opsin ingroup relationships or statements made.

Abstract:Please avoid the word "basal" to describe a living species (as in "larval eyes of a basal mollusk").

We deleted the word basal in the Abstract and corrected the wording in other sentences throughout the text.

"Transporters for opsin". To be precise you found homologs of proteins known in other species to be transporters.

Changed accordingly.

"today living animals" is not grammatical. You mean "extant animals".

Changed accordingly.

Following the suggestions for the Abstract (see also further down) increased word number above the limit of 150 words. To overcome this situation we changed the order and wording of sentences in the second half of the Abstract.

Introduction:"in times," is not grammatical. I think you can remove it.

Changed accordingly.

First paragraph; Photoreceptors are not the same as eyes. I suppose you might be discussing old debates, but these lines make it seem like PRC=eye.

We now clearly distinguish between PRCs and eyes.

Second paragraph; "upcoming" is the wrong word. You mean "Subsequent".

Changed accordingly.

Second paragraph; Evolution-based (needs hyphen). Remove the word "meanwhile", it is not grammatical as written.

Changed accordingly.

Second paragraph, last sentence; Not grammatical. This is a key sentence, where the authors are trying to state why a general reader should care about all the detail they are laying out. This important sentence needs to make sense, and be grammatical. (and probably should not be the last sentence of a paragraph).

We changed the sentence a bit to correct grammar and moved it and the subsequent sentence to the first paragraph of the Introduction.

Third paragraph; "enrolled" is wrong word. "are involved in", perhaps.

Changed accordingly.

Third paragraph; "In consequence". Not grammatical.

Changed accordingly.

Last paragraph; "independent evolutionary origin". Misleading. They are both opsins. What you mean is they are paralogs, not orthologs.

Changed accordingly.

Last paragraph; Living animals are not "basal".

Changed accordingly.

Last paragraph; What does "signaling mode" mean?

Changed to light transduction and molecular physiology

Results:"screening own" not grammatical

Changed accordingly.

Subsection “A new xenopsin from the chiton Leptochiton asellus”, first paragraph; "a few cnidops". The original name is "cnidops" not "cnidopsin". You also need the "a" to get across what you mean.

Both terms cnidopsin and cnidops are commonly used by the scientific community, but we agree that Plachetzky et al. (2007) first introduced the opsin subgroup as cnidops. Thus, we changed cnidopsins to cnidops throughout the manuscript.

"Since the distinction between xenopsins and c-opsins is so far only based on maximum- likelihood tree inference, we performed both maximum-likelihood and Bayesian analyses." This is not a logical statement. By this logic – you should also perform Parsimony, Neighbor Joining, and random tree generation.

The two most accurate and widely used methods for gene tree generation are maximum-likelihood and Bayesian inference. In many studies both methods are applied in parallel and results are checked for congruence. In contrast, maximum Parsimony and neighbour joining are rarely used due to well-known tree interference artifacts. Xenopsins were hitherto only recovered by maximum-likelihood methods. Thus, we were interested, whether xenopsins are also recovered by Bayesian inference.

We rewrote the sentence to make our intention more clear.

Furthermore, a lot of this is methods. Why is it in the Results section?

We describe in the Results section roughly the main steps of the performed analyses, which in our opinion are relevant to understand and to evaluate the significance of the chosen approach. The description is restricted to the choice of main parameters, special efforts in data mining (public raw RNA-seq data) and reasoning why certain sequences were excluded from subsequent analyses. All details are given in the Materials and methods section. We prefer to keep this section as it is, since in our experience this way of writing is quite usual in many papers dealing with extensive molecular phylogenetic analyses.

Subsection “A new xenopsin from the chiton Leptochiton asellus”, first paragraph; What special care?

We avoid this wording and rewrote the whole sentence.

Subsection “A new xenopsin from the chiton Leptochiton asellus”, first paragraph; "Own". Do you mean newly generated data?

Changed accordingly.

Subsection “A new xenopsin from the chiton Leptochiton asellus”, first paragraph; A living annelid is not basally branching.

Changed accordingly.

Subsection “A new xenopsin from the chiton Leptochiton asellus”, last paragraph; "paralogs distribute within and outside of a well-supported xenopsin subgroup". Not clear what that means.

We rewrote this sentence.

Subsection “Gene structure supports independent origins of xenopsins and c-opsins irrespective of their mutually exclusive distribution”, first paragraph; "are available" is the wrong word. "are recovered" – perhaps or "are identified" (if you want to stick with passive voice).

Changed accordingly.

Subsection “Gene structure supports independent origins of xenopsins and c-opsins irrespective of their mutually exclusive distribution”, first paragraph; The living annelid is still not basal.

Changed accordingly.

Subsection “Gene structure supports independent origins of xenopsins and c-opsins irrespective of their mutually exclusive distribution”, second paragraph”, last sentence; "anyway" is not usually used this way in written English.

Changed accordingly.

Subsection “Gene structure supports independent origins of xenopsins and c-opsins irrespective of their mutually exclusive distribution”, last paragraph” Again "independent" is misleading. They are both opsins, but different paralogs.

Changed accordingly.

Subsection “Gene structure supports independent origins of xenopsins and c-opsins irrespective of their mutually exclusive distribution”, last paragraph; "corroborate" -> "corroborating".

Changed accordingly.

Abstract: “Gene structure shows that xenopsin and ciliary opsins are of independent origin irrespective of their mutually exclusive distribution in animals.”Gene structures can evolve. Please tone down to:"Highly conserved but distinct gene structures suggest that xenopsins and ciliary opsins are of independent origin, irrespective of their mutually exclusive distribution in animals."

Changed accordingly.

Introduction: "By deeply mining public and new data we find xenopsins to be present in several taxa of protostome invertebrates."Please be more precise, otherwise you suggest a broader distribution than detected: "… to be present in several taxa of protostome lophotrochozoans."

We kept the original version, since we found xenopsins not only in the lophotrochozoan taxa mollusks, annelids and brachiopods, but also in platyhelminths and rotiferans, which both do not form part of Lophotrochozoa.

Results: "Several of the key players of ciliary opsin targeting, namely Las-Arf4, Las-rab8, Las^-1^94 FIP3, Las-RPGR and Las-MyosinVIIa". Again, it is more appropriate to refer to:"several general players of ciliary targeting also active in ciliary opsin targeting…"

Changed accordingly.

Discussion: I don't understand why you write "presence of xenopsin in ancestral eye PRCs seems likely". What do you mean by 'ancestral' here? Do you mean the ancestor of all bilaterians? Any ancestor (like of chitons)? Ancestor of all eyes.

Above this paragraph we discuss the possibility that the documented presence of xenopsin in eyes of a mollusc and of a brachiopod may be due to common ancestry. We mention both variants without clearly preferring one or the other: presence already in the lophotrochozoan or the bilaterian last common ancestor.

In the current paragraph being kind of a conclusion we likewise aim to not prefer one of these variants. We rewrote the sentence to provide a more precise wording.

Discussion:First sentence; Retina -> retinal.

Changed accordingly.

Third sentence; Grammar problem – as written, this sentence means only a few protostome sequences were known. You mean that only a few protostome c-opsins were known.

Changed accordingly.

Second paragraph; “Probably due to the increased taxon sampling…” and due to formal reconciled tree analysis.

Changed accordingly.

Third paragraph; basal.

Changed accordingly.

Fifth paragraph; Idea – the convergent evolution of the NKQ motif made xenopsin and c-opsins use the same phototransduction cascade. This could make them redundant, and allow for only one to usually be retained.

This is a very interesting suggestion. However, we regard (and write in the Discussion) that both Gi/cGMP and Gs and cAMP signalling as described from cnidops (many of which exhibiting motifs similar to NKQ) may be possible in the case of xenopsins. To make things not to complex, we prefer at the moment to leave the Discussion as it is.

Materials and methods:Subsection “Opsin tree inference”, second paragraph; Should cite Cannon et al. for Xenacoelomorpha.

Changed accordingly.

Subsection “Xenacoelomorpha”, second paragraph; But the papers you cite show that including distant GPCR outgroups is a problem, yet that is what you chose to do.

Certainly, outgroup composition is important in all kind of phylogenetic analyses. While inclusion of divergent outgroup members may distort tree topology by long branch attraction, very small and homogenous outgroups may result in a wrong rooting position. To prevent both problems, we decided to add some other GPCRs (octopamine, serotonin and adrenergic receptors) to the commonly used outgroup members melatonin receptors and Trichoplax opsin-like sequences. We do not regard the added sequences as being distantly related to opsins. They all belong to class α rhodopsin-like GPCRs (see Frederikson et al. (2003): Mol Pharm 63, Mickael et al. 2016: Comp Biochem Phys D 20), group rather close to opsins in Feuda et al. (2012) and are amongst the best blast and HMMER hits of melatonin receptors and Trichoplax opsin-like GPCRs. There is no evidence that these GPCR groups have a negative effect on opsin tree topology. The clear effect on ctenopsin position described in Feuda et al. (2014) was observed, when somatostatin receptors were included, which do not fall into the α group of rhodopsin-like GPCRS and indeed are very likely rather distantly related to opsins.

Figures:Why did you decide to root the tree with all the non-opsins as monophyletic? Others have found that Trichoplax opsin is a sister group to canonical opsins. Your tree is consistent with this, but you seemed to have chosen the outgroups to be monophyletic, with no justification as to why.

We changed tree rooting accordingly. See first comment.